# Since Faithfulness Fails: The Performance Limits of Neural Causal Discovery

**Mateusz Olko** [* 1 2]   **Mateusz Gajewski** [* 1 3]   **Joanna Wojciechowska** [2]
**Mikołaj Morzy** [3]   **Piotr Sankowski** [2 4 5]   **Piotr Miłoś** [2 1 6 7]

## Abstract

Neural causal discovery methods have recently improved in terms of scalability and computational efficiency. However, our systematic evaluation highlights significant room for improvement in their accuracy when uncovering causal structures. We identify a fundamental limitation: *unavoidable likelihood score estimation errors disallow distinguishing the true structure*, even for small graphs and relatively large sample sizes. Furthermore, we identify the faithfulness property as a critical bottleneck: (i) it is likely to be violated across any reasonable dataset size range, and (ii) its violation directly undermines the performance of neural penalized-likelihood discovery methods. These findings lead us to conclude that progress within the current paradigm is fundamentally constrained, necessitating a paradigm shift in this domain.

## 1. Introduction

Causal discovery is essential for scientific research, driving a growing demand for machine learning methods to support this process. Despite the development of several neural causal discovery methods in recent years (Brouillard et al., 2020; Lorch et al., 2021; Annadani et al., 2023; Nazaret et al., 2024), their performance remains insufficient for real-world applications, particularly in fields such as medicine and biology (de Castro et al., 2019; Peters et al., 2016). Furthermore, these methods are usually evaluated using datasets, which vary between studies, obscuring the overall picture and making the assessment of advances difficult.

In response to this challenge, we introduce a unified benchmarking protocol for neural causal discovery methods. Specifically, we use identical datasets, tune hyperparameters consistently, and use a standardized functional approximation across all methods. Through this systematic evaluation, we uncover a concerningly low level of performance. This raises a fundamental question: Why do these methods struggle to recover the ground-though causal structure even when the data is generated synthetically with all theoretical assumptions fulfilled and the dataset is large?

We identify a fundamental limitation: unavoidable likelihood estimation errors disallow distinguishing the true structure. This happens primarily due to finite size of the datasets. However, we observe minimal improvements when scaling either the data or the network size. At a technical level, we pinpoint the faithfulness property as a critical bottleneck. We formalize faithfulness violations in finite datasets using the strong-faithfulness measure. Then, we show that any strength of the faithfulness is easily violated when graph size or density are increased. Even for moderate size graphs it is likely to be violated across any reasonable dataset size range. We believe these limitations highlight the need for a paradigm shift rather than incremental improvements in architecture or optimization techniques.

In more details, our contributions are as follows:

1. In a carefully controlled experiment, we demonstrate that the likelihood loss, as used by causal discovery approaches, faces estimation errors which disallow the recovery of ground-truth causal graphs, even for small graphs and relatively large sample sizes.

2. We propose practical method of approximation of $\lambda$-strong faithfulness property for nonlinear datasets. We experimentally demonstrate how proportion of $\lambda$-unfaithfull distributions scales with the size and density of graphs from Erdos-Renyi class.

3. We provide results of unified benchmark for neural causal discovery methods and show that their performance correlates with $\lambda$-strong faithfulness property of the datasets.

---

[*]Equal contribution   [1]IDEAS NCBR, Warsaw, Poland [2]University of Warsaw, Warsaw, Poland [3]Poznan University of Technology, Poznan, Poland [4]MIM Solutions, Warsaw, Poland [5]Research Institute IDEAS, Warsaw, Poland [6]Institute of Mathematics, Polish Academy of Sciences, Warsaw, Poland [7]deepsense.ai, Warsaw, Poland. Correspondence to: Mateusz Olko <mateusz.olko@gmail.com>, Mateusz Gajewski <mg96272@gmail.com>.

*Proceedings of the 42nd International Conference on Machine Learning*, Vancouver, Canada. PMLR 267, 2025. Copyright 2025 by the author(s).

## 2. Background and Related Work

**Structural causal models (SCMs).** Causal relationships are commonly formalized (Pearl, 2009) using SCMs, which represent causal dependencies through a set of structural equations. For a directed acyclic graph (DAG) $G = (V, E)$, an SCM is defined by a set of equations of the form:

$$X_i = f_i(Pa_G(i), U_i), \qquad (1)$$

where $X_i$ is a random variable at vertex $i \in V$, $f_i \colon \mathbb{R}^{|Pa_i|+1} \to \mathbb{R}$ is a function, $Pa_G(i)$ denotes the set of parents of $i$ in $G$, and $U_i$ is an independent noise. In this work, we assume *additive noise* SCMs, also referred to as *additive noise models* (ANM), viz.,

$$f_i(Pa_G(i), U_i) = g_i(Pa_G(i)) + U_i \qquad (2)$$

for some $g_i \colon \mathbb{R}^{|Pa_G(i)|} \to \mathbb{R}$. An SCM $\mathcal{M} = (G, \{f_i\}_{i \in V}, \{U_i\}_{i \in V})$ defines a joint distibution $P$ over the set of random variables $\{X_i\}_{i \in V}$.

**Causal discovery.** Causal discovery aims to uncover causal structure $G$ of a SCM based on data sampled from the joint distribution $P$. However, in general the solution can only be identified up to a Markov Equivalence Class (MEC), the set of DAGs encoding the same conditional independencies (Verma & Pearl, 1990).

While there is a line of classical methods like PC (Spirtes et al., 2000) or GES (Chickering, 2020), those algorithms work well for linear relationships and moderate-sized problems, they struggle in more challenging scenarios. Neural causal discovery methods have emerged as a promising direction, offering class of methods that could potentially handle nonlinear relationships and provide computational efficiency, and scalability in both data size and number of variables.

Evaluating causal discovery methods presents unique challenges. In real-world applications, ground truth causal structures are typically unknown or can only be partially elicited from domain experts with inherent noise and bias. This fundamental limitation necessitates the use of synthetic data for evaluations, as they provide controlled environments essential for understanding the methods' capabilities and limitations.

**Recent developments in neural causal discovery.** We would like to highlight four recent approaches to neural causal discovery: DCDI (Brouillard et al., 2020), SDCD (Nazaret et al., 2024), DiBS (Lorch et al., 2021), and BayesDAG (Annadani et al., 2023), which in our view effectively represent the major developments in neural causal discovery from past years. DCDI represents the evolution of NO-TEARS-based methods like GRAN-DAG (Zheng et al., 2018; Lachapelle et al., 2019), improving upon the original by separating structural and functional parameters and incorporating interventional data. SDCD unifies and advances various acyclicity constraints proposed in methods like NO-BEARS (Lee et al., 2019) and DAGMA (Bello et al., 2022), demonstrating superior performance compared to SCORE (Rolland et al., 2022) and DCDFG (Lopez et al., 2022). For Bayesian approaches, we chose DiBS, which incorporates NO-TEARS regularization in its prior, and BayesDAG, which builds on NO-CURL's DAG parametrization (Yu et al., 2021) using MCMC optimization.

All these approaches use a continuous representation of the graph structure, enforcing a differentiable acyclicity constraint to ensure the result is a DAG. The primary objective is to maximize $\log p_\theta(X|\mathcal{G})$, that is the log-likelihood of the data given the graph while incorporating regularization terms to control graph complexity. Such methods are in general described as penalized likelihood causal discovery methods. The discovery procedure comprises two parts: fitting functional approximators and structure search, which are usually done in parallel.

The penalized likelihood methods, are among most general class of methods. In principle they do not need any parametric assumptions on functions and noises. DCDI (Brouillard et al., 2020) and SDCD (Nazaret et al., 2024) can work in non-linear setting with non-additive noise, while Bayes-Dag (Annadani et al., 2023) and DiBS (Lorch et al., 2021) make only additive noise assumption.

**Hardness of causal discovery.** The so-called faithfulness property is a fundamental assumption commonly used by causal discovery methods (Pearl, 2009; Brouillard et al., 2020). It is formulated as follows:

$$\forall_{a,b \in V} \forall_{S \subseteq V \setminus \{a,b\}} X_a \perp\!\!\!\perp X_b | X_S \iff a \perp_{\mathcal{G}} b | S. \quad (3)$$

where $\perp\!\!\!\perp$ denotes conditional independence and $\perp_{\mathcal{G}}$ denotes $d$-separation. For more information on $d$-separation please refer to Appendix A.1. The faithfulness property can be violated when multiple causal paths cancel each other (see example in Appendix A.2). While the set of unfaithful distributions associated with a given DAG has measure zero (Boeken et al., 2025), distributions may exhibit arbitrarily weak conditional dependencies. When dealing with final datasets, such weak dependencies become indistinguishable from independence.

This observation motivated Zhang & Spirtes, 2003 to introduce the $\lambda$-strong faithfulness assumption. A distribution $P$ is said to be $\lambda$-strong faithful to a DAG $\mathcal{G}$ if:

$$\forall_{a,b \in V} \forall_{S \subseteq V \setminus \{a,b\}} |\rho_P(X_a, X_b | X_S)| > \lambda \iff a \not\perp_{\mathcal{G}} b | S, \quad (4)$$

where $\rho_P(X_a, X_b|X_S)$ denotes a partial correlation coefficient and $\lambda \in (0, 1)$[1]. For linear systems, this assumption ensures uniform consistency of the PC algorithm with $\lambda \propto 1/\sqrt{n}$ where the sample size $n \to \infty$. Therefore, $\lambda$ can serve as a notion of the difficulty of the causal discovery task.

Notably Uhler et al. (2013) proved that, in case of linear SCMs, for any fixed $\lambda > 0$, the fraction of $\lambda$-strong faithful distributions decreases exponentially with graph size and density, suggesting fundamental limitations in causal discovery on large graphs. The work provides theoretical results for linear SCMs and the PC algorithm.

However so far little has been shown regarding nonlinear functions and contemporary neural network approaches. We aim to breach the gap with our experimental contributions, thus the remainder of the paper is structured as follows. First, we build a framework that allows us to measure the difficulty of the problem, expressed by $\lambda$-strong faithfulness property, and show it grows quickly with the size and density of the underlying graph (Section 3.1). Than, we demonstrate that the difficulty is connected to the number of samples required to recover the true graph (Section 3.2). Our framework mimics the linear case described above. Finally, in Section 4, we evaluate contemporary neural causal discovery methods and analyze their performance in relation to the problem difficulty measure.

## 3. Estimation of Strong Faithfulness Property and Likelihood Score

For a fixed distribution $P$ and associated with it graph $\mathcal{G}$ there can be multiple $\lambda$ that satisfy the Equation 4. Therefore, in the remainder of the paper we will denote $\lambda$ as the maximal threshold satisfying the equation. Additionally, as we are working with nonlinear data, we use Spearman correlation coefficient, which we will denote $\rho_P$ with a slight abuse of notation. Specifically, for a given distribution $P$ associated with graphs $\mathcal{G}$ we define:

$$\lambda = \max\{t : \forall_{a,b \in V} \forall_{S \subseteq V \setminus \{a,b\}}$$
$$|\rho_P(X_a, X_b|X_S)| > t \iff a \not\perp_{\mathcal{G}} b|S\}. \quad (5)$$

### 3.1. $\lambda$ decreases quickly with growing graph size and density.

We aim to measure the difficulty of causal discovery given non-linear dataset $\mathcal{D}$ and associated with it structure $\mathcal{G}$ expressed by the $\lambda$ parameter. Theoretically, $\lambda$ could be determined by computing partial correlations for all variable pairs across all conditioning sets and identifying a threshold

---

[1] When $\lambda = 0$, $\lambda$ this reduces to the standard faithfulness assumption.

that separates conditionally $d$-separated from $d$-connected nodes (see equation 5). In practice, due to irreducible errors in correlation estimation from finite data and the presence of small true correlation values, it is infeasible to establish the separating threshold. Instead we choose the threshold that maximizes the F1-score of classification to $d$-separated and $d$-connected nodes.

**Effective approximation of $\lambda$.** Specifically, we use partial Spearman correlation, classifying node pairs as $d$-separated if their conditional correlation coefficient, computed from the finite dataset, falls below a given threshold. We then define $\hat{\lambda}$ as the threshold that optimizes the F1-score of this classification. Formally, for dataset $\mathcal{D}$ and associated with it DAG $\mathcal{G}$:

$$\hat{\lambda} = \operatorname{argmax}_t F_1(t, \mathcal{D}, \mathcal{G}), \quad (6)$$

$$F_1(t, \mathcal{D}, \mathcal{G}) = 2 \cdot \frac{\text{TP}(t, \mathcal{D}, \mathcal{G})}{\text{P}(t, \mathcal{D}, \mathcal{G}) + \hat{\text{P}}(t, \mathcal{D}, \mathcal{G})}$$

$$\text{P}(t, \mathcal{D}, \mathcal{G}) = \#\{a, b, S : a \perp_G b|S\}$$
$$\hat{\text{P}}(t, \mathcal{D}, \mathcal{G}) = \#\{a, b, S : |\rho_D(a, b|S)| < t\}$$
$$\text{TP}(t, \mathcal{D}, \mathcal{G}) = \#\{a, b, S : |\rho_D(a, b|S)| < t \text{ and } a \perp_G b|S\}$$

where $\rho_D(a, b|S)$ denotes the partial Spearman correlation coefficient computed from dataset $\mathcal{D}$, and $a \perp_G b|S$ denotes $d$-separation in graph $\mathcal{G}$. Note, that when data size increases estimation errors in the partial Spearman correlation $\rho_D(a, b|S)$ decrease. Thus, in the limit of the data $\hat{\lambda}$ converges to $\lambda$. For an experimental evaluation of the estimation accuracy and precision on finite datasets, please refer to Appendix D.

**Fraction of $\lambda$-strong distributions.** In case of linear datasets, the relation between the fraction of sampled $\lambda$-strong distribution and the graph density and number of nodes has been extensively studied by Uhler et al. (2013). It was proved, for several types of connected graphs, that the fraction of $\lambda$-strong faithful distributions decreases exponentially. Here we provide analogous experimental results for nonlinear SCMs based on connected Erdos-Renyi graphs. The details of the experimental setup and results for more graph types are provided in the Appendix B.

In Figures 1a and 1b we plot how fraction of $\lambda$-unfaithful distributions changes with the number of nodes and the density of the graph respectively. Our experimental results align with theoretical study on linear datasets. We observe that for a fixed $\lambda$ and number of nodes, the proportion of distributions that are not $\lambda$-strong decreases with the expected neighborhood size, similar observation can be made when we increase the number of nodes in a graph. Even for relatively small and sparse graphs (6 nodes and expected

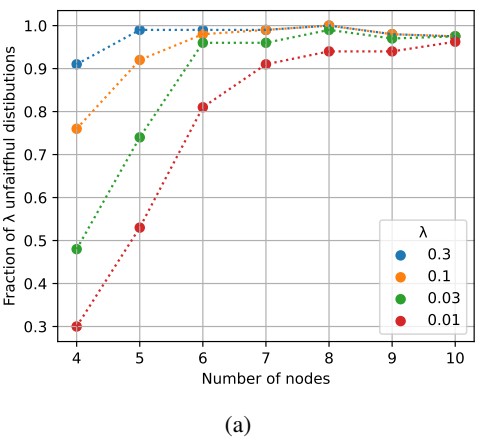

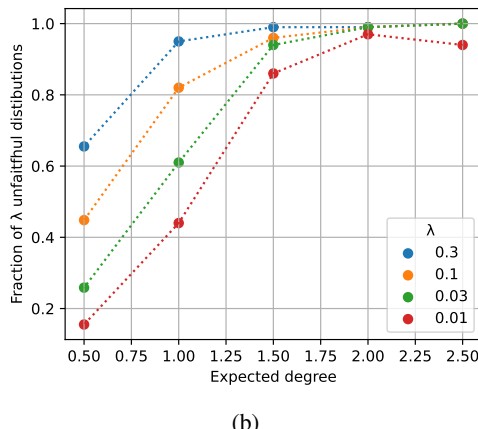

Figure 1: (a) Estimated fraction of $\lambda$-unfaithful distributions for Erdos-Renyi graphs with various number of nodes. (b) Estimated fraction of $\lambda$-unfaithful distributions for Erdos-Renyi graphs with 6 nodes and varying density. Colored lines correspond to specific values of $\lambda$.

neighborhood size 1) more than $60\%$ of distributions are not faithful for $\lambda = 0.03$ and more than $80\%$ for $\lambda = 0.1$. We get similar results for bipartite, small-world, and scale-free graphs. They are presented in Figure 7 in Appendix B.

### 3.2. Lower $\hat{\lambda}$ requires larger sample size

We investigate how $\lambda$ affects convergence rate of causal discovery algorithms with neural-network approximators. To be able to draw conclusions about impact of data sample on the accuracy of causal discovery we design a conceptually simple neural network based causal discovery method that minimizes approximation errors. We evaluate it in a controlled setting that enables a comprehensive analysis to identify bottlenecks in the neural causal discovery process.

**Exhaustive estimation approach.** Our goal is to estimate the regularized log-likelihood score, a common objective function in neural causal discovery. For a graph $G$ with nodes set $V$ the log-likelihood of the dataset $D = \{X_i\}_{i \in V}$ can be expressed as the sum of log-likelihoods of individual vectors $X_i$:

$$L(G) = \sum_{i \in V} - \log p(X_i | \mathrm{Pa}_G(i)). \quad (7)$$

Since, we need to be able to compute the score for each graph $G$ in class of DAGs over the node set $V$, we train a neural-network-based density function $f(X_i; X_{\mathrm{Pa}_G(i)}; \theta_{i,\mathrm{Pa}_G(i)})$ for each variable $X_i$ and each parent set $\mathrm{Pa}_G(i) \subseteq V \setminus \{i\}$. Note, that unlike scalable NN approaches, we have a different set of parameters $\theta_{i,Pa_{(G)}(i)}$ for each parent set $\mathrm{Pa}_G(i)$. This reduces approximation errors. The penalized score can now be written as:

$$S(G) = \sum_{i \in V} - \log f(X_i; X_{\mathrm{Pa}_G(i)}; \theta_{i,\mathrm{Pa}_G(i)}) + \gamma |G| \quad (8)$$

where $|G|$ denotes number of edges in $G$ and $\gamma > 0$ is a sparsity coefficient.

Further, to reduce approximation errors due to initialization, we also employ random seed bootstrapping. The training procedure is repeated $N$ times starting from different random initializations. The final formula for the score uses the average of $N$ estimates:

$$\hat{S}(G) = \sum_{i \in V} - \frac{1}{N} \sum_{n=1}^{N} \log f(X_i; X_{\mathrm{Pa}_G(i)}; \theta^n_{i,\mathrm{Pa}_G(i)}) + \gamma |G|. \quad (9)$$

This procedure drives the approximation error down to a negligible level, at the cost of significant computational effort, and, through bootstrap confidence intervals, lets us isolate estimation error from approximation error.

We summarize our efforts to minimize approximation error:

1. The neural network approximator and the data-generating model belong to the same class – no model misspecification.

2. Additionally, we evaluate multiple network architectures and confirm our architectural choice experimentally (see Table 3 in Appendix C).

3. We use an ensemble approach over multiple random initialization, which not only improves accuracy but also provides bootstrap confidence interval for the approximation error.

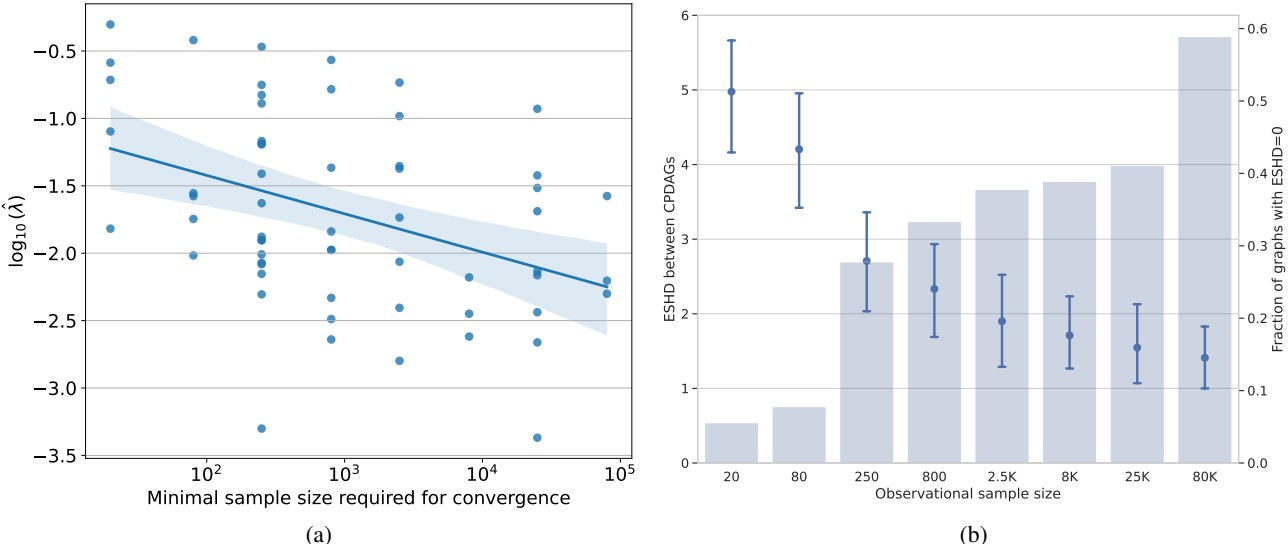

(a)                                                                 (b)

Figure 2: (a) Relation of sample needed to for a distribution to converge and the $\hat{\lambda}$. (a) Linear regression fit between the minimal sample size required for true structure recovery and $\log_{10}(\hat{\lambda})$ measure. The spearman rank correlation of those measures is $-0.93$ and p-value for correlation is 9e-4. (b) Comparison of the performance of NN-opt method depending on data size. Averaged over 90 samples. For definition of ESHD$_{\text{CPDAG}}$ please refer to Section 4.

Assuming $\gamma$ is small enough and the faithfulness assumption holds, the DAG that minimizes the score in Equation 8 will belong to the MEC of data generating graph (Brouillard et al., 2020).

*Remark* 3.1. Notice that, given score estimation described above, we can discover the MEC by exhaustive evaluation of all DAGs in the given class. The method is not practically useful, but can illustrate the influence of estimation and approximation errors on causal discovery process. Since, graph search component is removed the method can also be viewed as upper-bound on existing differentiable approaches to causal discovery relying on similar neural-network approximators. The pseudocode of the method can be found in Appendix G.

**Synthetics causal data.** We generate synthetic data with a known ground-truth causal structure. We consider causal DAGs with only five nodes $V = \{1, \ldots, 5\}$. We generate these DAGs using the Erdos-Renyi model with the expected number of 5 edges. The functional relations between nodes are modeled by randomly initialized MLPs with two hidden layers of size 8. Additionally, we use additive Gaussian noise: $U_i \sim \mathcal{N}(0, \sigma_i)$, where $\sigma_i$ may depend on $i \in V$. For more details refer to Appendix E.1.

**The maximum likelihood estimator is biased.** The MLE objective yields biased estimates of the score. This bias is not systematic; instead, it varies randomly depending on the training sample. We illustrate exemplary score es-

timates in Figure 3. For this experiment, we generated a dataset as described above and applied our exhaustive estimation method using 800 samples. We used a bootstrap sample size of $N = 29$. If the score estimates were accurate, the structures within the Markov Equivalence Class (MEC) would consistently receive the lowest scores (highlighted in red in the figure). However, we observe that a substantial number of alternative structures are assigned comparable or better scores – depicted in blue and green, respectively. Importantly, when the dataset size is increased to 8,000, the number of green and blue structures decreases significantly, indicating that the exact faithfulness assumption holds.

**The convergence rate of our causal discovery method correlates with $\hat{\lambda}$.** We generate 90 synthetic datasets and evaluate causal discovery approach described in Remark 3.1. We experiment with various neural-network sizes and select the best performing one which happens to be identical as used to generate the data (see Appendix C). The ensemble size used in this experiment is $N = 3$. For each dataset we evaluate causal discovery on subsets of varying sizes. The size of the smallest subset required for convergence is recorded and compared to value of $\hat{\lambda}$ of the dataset. The results are presented in Figure 2a. The experiment demonstrates that datasets with higher $\hat{\lambda}$ require usually fewer samples for successful recovery. The $\hat{\lambda}$ and minimal number of samples is highly correlated, the spearman correlation coefficient is $-0.93$.

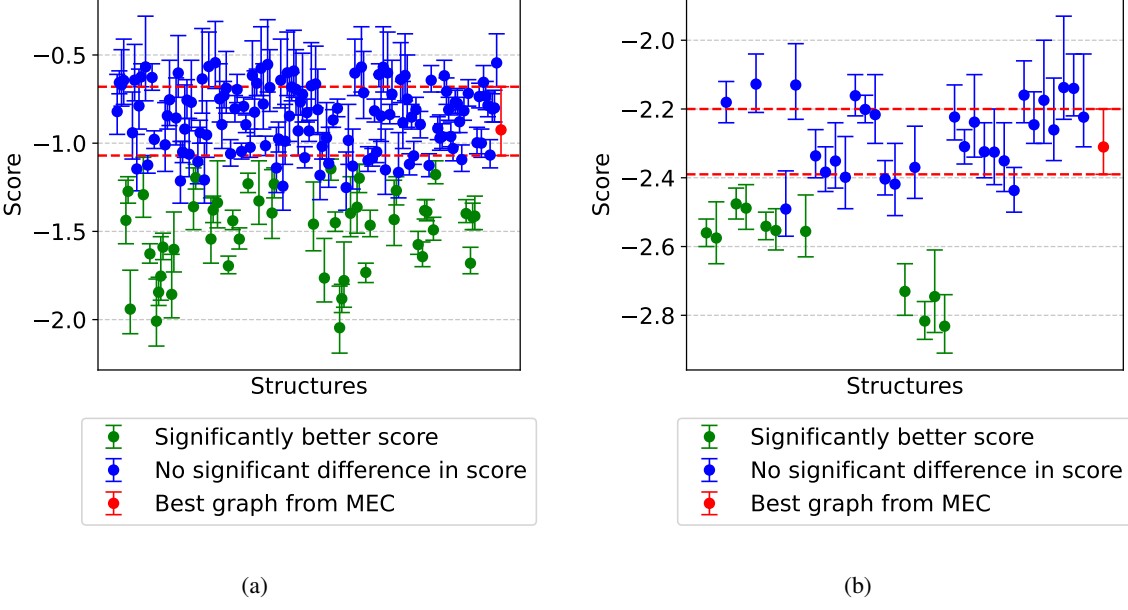

(a)                                                                    (b)

Figure 3: Exemplary results of score evaluation using our robust neural network based approximation approach. In red —
score of the target structure, in green — scores of structures with statistically significantly better scores, in blue — scores
of structures with comparable scores. The error bars signify 95% confidence intervals. Note that considerable number of
structures has significantly better (lower) score that the target structure. (a) Plot for 800 samples. (b) Plot for 8000 samples.

**Accuracy of causal discovery improves with data size.**
Apart from convergence, we want to quantify structural errors when recovery is incomplete, to assess model reliability and compare methods. Understanding how these errors decrease with more data provides insight into the efficiency of the learning process. The results are presented in Figure 2b. For very small datasets we observe a relatively big $ESHD_{CPDAG}$ of 4, which rapidly improves with sample size. As the sample size grows, the structure discovery accuracy stabilizes. For sample sizes of 2,500 and 8,000, the average value of $ESHD_{CPDAG}$ is just below 2. In the dataset used for this experiment, the average number of edges in CPDAG is around $8.4$, meaning that on average almost 25% of the edges are predicted incorrectly.

As previously observed, datasets vary in difficulty, and the number of correctly identified structures increases with sample size. While the average $ESHD_{CPDAG}$ remains significantly above zero, structural recovery improves notably from 6 converged graphs at 80 samples to 26 at 250. This trend continues, though at a diminishing rate: 34 and 35 graphs converge at 2,500 and 8,000 samples, respectively, and 37 and 53 at 25,000 and 80,000. These results indicate that larger sample sizes enable more accurate structure discovery. However, the marginal gains decrease, and even with 80,000 samples, correct recovery of all structures is not achieved.

### 3.3. Conclusions

In this section, we provided a detailed experimental analysis of the relationship between the $\lambda$-strong faithfulness property of non-linear datasets and the performance of causal discovery with neural function approximators. Our results indicate that the $\lambda$-strong faithfulness property constrains causal discovery from finite data, limiting both convergence rate and accuracy. Furthermore, we demonstrate in a controlled setting that data scarcity introduces significant errors in score assignment. Additionally, we show that the proportion of $\lambda$-strong faithful distributions in ER graphs with nonlinear functions rapidly decreases as graph density and size increase. Those findings, along with the theoretical results of Uhler et al. (2013), suggests that current neural causal discovery methods may face fundamental limitations.

## 4. Evaluation of Neural Causal Discovery Methods

After linking the challenges in neural causal discovery to $\lambda$ - strong faithfulness in Sections 3.2 and 3.1, in this section, we investigate how this violation affects neural causal discovery methods performance in terms of structural metrics. Additionally since these methods were constructed with scalability in mind, the experiments can be performed on graph bigger that in Section 3.

| | ER(5, 1) | | ER(10, 2) | | ER(30, 2) | |
|---|---|---|---|---|---|---|
| Method | ESHD$_{\text{CPDAG}}$ | F1-Score$_{\text{CPDAG}}$ | ESHD$_{\text{CPDAG}}$ | F1-Score$_{\text{CPDAG}}$ | ESHD$_{\text{CPDAG}}$ | F1-Score$_{\text{CPDAG}}$ |
| DCDI | 5.7 (3.7, 8.1) | 0.60 (0.46, 0.74) | **16.9** (15.7, 18.1) | 0.52 (0.50, 0.56) | 45.9 (42.0, 49.9) | **0.73** (0.69, 0.77) |
| BayesDAG | 3.9 (3.6, 4.3) | 0.78 (0.77, 0.81) | 18.3 (16.9, 19.8) | **0.56** (0.54, 0.59) | 51.7 (48.2, 55.9) | 0.59 (0.57, 0.61) |
| DiBS | **2.6** (1.7, 3.7) | **0.85** (0.80, 0.90) | **16.9** (14.2, 201) | **0.61** (0.57, 0.68) | 68.0 (65.3, 70.9) | 0.23 (0.22, 0.24) |
| SDCD | 5.4 (3.8, 6.7) | 0.60 (0.35, 0.69 | 20.9 (19.5, 22.2) | 0.54 (0.46, .62) | 62.8 (58.8, 67.7) | 0.55 (0.53, 0.58) |

Table 1: Comparison of ESHD$_{\text{CPDAG}}$ and F1-Score$_{\text{CPDAG}}$ for different methods on ER(10, 2) (left) and ER(30, 2) (right) dataset. The numbers in the subscripts correspond to 95% confidence intervals. The statistics were computed based on 30 graphs.

## 4.1. The systematic benchmarking protocol

We evaluate methods DiBS, DCDI, BayesDAG, and SDCD introduced in Section 2 on identical datasets, tune hyperparameters consistently, and use a common functional approximation.

**Dataset generation**   We sample three types of graphs from the Erdős-Rényi (ER) distribution (Erdös & Rényi, 1959) as described in 3.1: one with 5 nodes and the expected degree of 1, another with 10 nodes and the expected degree of 2, and the third with 30 nodes and the expected degree of 2. These datasets are referred to as ER(5, 1), ER(10, 2), and ER(30, 2), respectively. These parameter choices align with commonly studied medium-sized graphs in causal discovery research (Brouillard et al., 2020; Nazaret et al., 2024).

**Hyperparameter tuning**   To ensure a fair comparison across all methods, we perform systematic hyperparameter tuning, selecting the best-performing parameters for each method We employ a grid search approach based on the parameter ranges suggested by the original authors. This process optimizes key variables such as regularization coefficients, sparsity controls, and kernel configurations. Details can be found in Appendix E.2.

**Functional approximators**   We standardize the choice of functional approximators across all experiments, using a two-layer MLP with a hidden dimension of 4 to model each functional dependence $f$ (see Section 2). This model size is consistent with previous work (Brouillard et al., 2020; Nazaret et al., 2024) and has proven to perform well across all the benchmarked methods, as discussed in Appendix E.3. Additionally, we use trainable variance to allow the model to adapt to varying noise levels, in line with our dataset generation setup.

**Structure evaluation**   We evaluate graph discovery within the MEC using ESHD$_{\text{CPDAG}}$ and F1-Score$_{\text{CPDAG}}$, where ESHD$_{\text{CPDAG}} = 0$ and F1-Score$_{\text{CPDAG}} = 1$ when the predicted graph is in the same MEC as the ground truth. For Bayesian methods, we computed the expected value by sampling from the posterior distribution; for non-Bayesian methods, we use a single graph.

The Structural Hamming Distance (SHD) (Tsamardinos et al., 2006) counts edge insertions, deletions, and reversals needed to match the predicted graph to the true graph. We define **Expected SHD between CPDAGs** as:

$$\text{ESHD}_{\text{CPDAG}}(\mathcal{G}, \mathbb{G}) =$$
$$= \mathbb{E}_{\mathcal{G}^* \sim \mathbb{G}}[\text{SHD}(\text{CPDAG}(\mathcal{G}), \text{CPDAG}(\mathcal{G}^*))], \quad (10)$$

where $\mathbb{G}$ is the resulting distribution of graphs, $\mathcal{G}^*$ is a graph sampled from $\mathbb{G}$ and $\mathcal{G}$ is the ground true graph. The F1-Score measures the harmonic mean of precision and recall for edge predictions. We compute the **Expected F1-Score between the CPDAGs** as follows:

$$\text{F1-Score}_{\text{CPDAG}}(\mathcal{G}, \mathbb{G}) =$$
$$= \mathbb{E}_{\mathcal{G}^* \sim \mathbb{G}}[\text{F1-Score}(\text{CPDAG}(\mathcal{G}), \text{CPDAG}(\mathcal{G}^*))]. \quad (11)$$

For more details and justification on the selection of metrics please refer to Appendix F.

**Methods comparison**   Table 1, summarizes the benchmark results of neural-based causal discovery methods on graphs from ER(5, 1), ER(10, 2), and ER(30, 2) classes. We tune hyperparameters to optimize the ESHD$_{\text{CPDAG}}$ metric. For all classes of graphs, metrics were computed based on 30 graphs.

The results show that DiBS is particularly effective for smaller graphs (ER(5, 1) and ER(10, 2)), while DCDI is able to achieve the best results for moderate-size graphs (ER(10, 2) and ER(30, 2)). The ranking of the methods changes with the size of the graphs but SDCD consistently exhibits the worst performance in terms of ESHD$_{\text{CPDAG}}$.

Nevertheless, the structural metrics of all the methods remains unsatisfactory with all methods predicting more than half of the edges incorrectly.

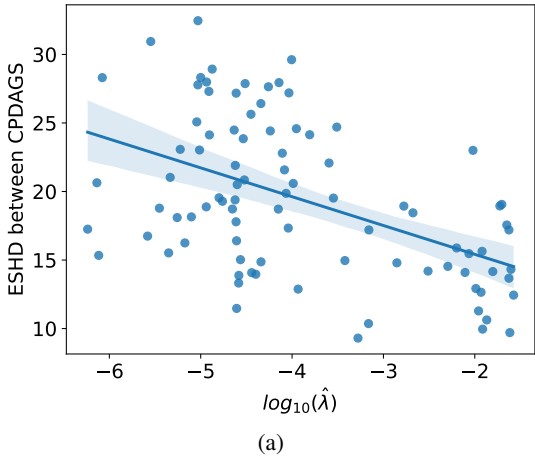

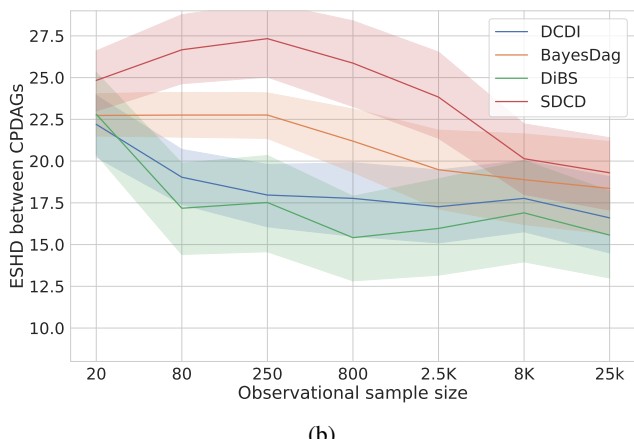

(a)                              (b)

Figure 4: (a) Linear regression fit between the average performance of neural causal discovery methods and $\log_{10}(\hat{\lambda})$ measure. The p-value for spearman rank correlation between $\hat{\lambda}$ and ESHD$_{\text{CPDAG}}$ and is 4e-6, signifying anti-monotonic correlation. b) Performance of benchmarked methods in terms of ESHD$_{\text{CPDAG}}$ with resect to dataset size for ER(10, 2) graphs, averaged over 30 samples.

## 4.2. $\hat{\lambda}$ and NCD method performance

To investigate the tie between $\hat{\lambda}$ metric and the difficulty of neural causal discovery, we generate 30 graphs from the ER(10, 2) class, introduced in Section 4.1. Based on each graph, we define three different SCMs, resulting in 90 distinct distributions. From each one we generate 8,000 observational samples. We then evaluate the $\hat{\lambda}$ of each dataset and compute the performance of the selected neural-based causal discovery methods.

In Figure 4a we present the relationship between average performance of all methods and the logarithm of $\hat{\lambda}$ (for readibility) for all 90 distributions in the dataset. The performance is better (lower SHD) for distributions with higher $\hat{\lambda}$. The Spearman's rank correlation coefficient is $\rho = -0.46$ and p-value for sperman correlation test is $4e - 6$. This result proves the strong anti-monotonicity between $\hat{\lambda}$ and average methods' performance.

## 4.3. Lack of scalability

We want to investigate the impact that the sample size has on a causal discovery methods. We expect it to behave similar to our findings from Section 3.2 as we hypothesize that the lack of performance is due to relation between data and approximators. We would like to note that the NCD methods use one neural network to approximate the distribution under changing set of parents, which is harder task than the one in Section 3.2. We evaluate the benchmarked methods on datasets with varying number of observational samples, ranging from 20 to 8,000 observations.

The results, presented in Figure 4b, reveal no consistent pattern of improvement in the ESHD$_{\text{CPDAG}}$ metric as observational sample size increases, despite extensive hyperpa-

rameter tuning (as described in Section 4.1). For example, DiBS shows the best performance on larger datasets, but its improvements plateau after around 800 samples. Similarly, BayesDAG shows only marginal improvements with larger sample sizes and is unable to outperform DiBS. DCDI improves up to 250 samples and then maintains consistent performance regardless of the sample size, similar to DiBS. Interestingly, SDCD's performance is poor on datasets with small number of observations but begins to improve once sample sizes exceed 250, though is unable to reach DCDI's performance, for larger sample sizes the rate of improvement decreases.

Further analysis of the effect of sample size on smaller graphs ER(5, 1)is presented in Figure 14 in Appendix F. Overall, the results on smaller graphs align with the trends observed on larger graphs. Specifically, while some methods improve with increasing sample size, others show inconsistent or even degraded performance.

Even for the bigger sample sizes the results remains unsatisfactory. This confirms our observations, made in the previous sections about datasets difficulty and the amount of samples required for convergence. We conclude, that the amount of data needed to recover a graph from a true MEC with those methods is out of scope for most applications.

## 5. Limitations and Future Work

- Work of Lippe et al. (2022) suggests that interventional data can replace the need for faithfulness assumption. A valuable extension of our research would be to evaluate the performance of the benchmarked methods on interventional datasets to understand their limitations and potential improvements in this context.

- Our work provides experimental evidence for the challenging nature of causal discovery from non-linear data, especially when using neural networks. It would be beneficial for the community to establish theoretical bounds on the best possible performance in such cases.

- Neural networks are well known approximators, with well established position in Machine Learning community. It will be however interesting to investigate other classes of aproximators, that could possibly have a better suited characteristic for the task of causal discovery.

- Our conclusions are based on characteristic of distributions associated with Bayesian Networks provided by Uhler et al. (2013). The result says that for any fixed $\lambda$, $\lambda$-strong faithful distributions vanish exponentially with the size of the graph. It may hold, though is highly unprobable, that real-world distributions adhere to $\lambda$-strong faithfulness despite large sizes of the graph. Further investigation is required.

## 6. Discussion

In this work, we conducted a detailed experimental analysis of the relationship between the $\lambda$-strong faithfulness property in nonlinear datasets and the performance of causal discovery methods based on neural function approximators. Our findings indicate that the $\lambda$-strong faithfulness assumption imposes a significant constraint on causal discovery from finite data, adversely affecting both convergence rates and accuracy.

We further demonstrate that the proportion of $\lambda$-strongly faithful distributions in various connected graph types with nonlinear mechanisms decreases rapidly as graph density and size increase. This phenomenon implies that, in practical scenarios involving large or dense graphs, the probability of encountering $\lambda$-faithful data becomes vanishingly small. We also show a clear correlation between the degree of $\lambda$-strong faithfulness and the performance of differentiable, score-based causal discovery methods, indicating that the assumption plays a central role in the tractability of these approaches.

To the best of our knowledge, this is the first study to empirically connect $\lambda$-strong faithfulness to the performance of score-based causal discovery methods, extending prior theoretical insights which focused primarily on independence-test-based approaches (Zhang & Spirtes, 2003). Our results suggest that gains in efficiency and accuracy for neural-based causal discovery methods may be approaching fundamental limits, driven by the intrinsic properties of the data distributions rather than algorithmic shortcomings.

These findings highlight the importance of evaluating causal discovery algorithms on real-world datasets, which may deviate significantly from synthetic benchmarks based on uniformly initialized neural networks. A promising direction in this regard is the recent work by Gamella et al. (2025), which uses physical simulations to generate causal datasets. This approach preserves the complexity and realism of physical systems while offering ground-truth causal structures, making it a valuable resource for future benchmarking.

While many causal discovery methods rely on the strong faithfulness assumption, alternative frameworks have been proposed. For linear structural causal models (SCMs), Van de Geer & Bühlmann (2013) demonstrated that sparsity-based assumptions—specifically the sparsest Markov representation—can recover causal structure without relying on strong faithfulness. Similarly, Ng et al. (2021) proposed a method that operates under a relaxed faithfulness condition, requiring fewer conditional independencies to hold. Another underexplored direction involves relaxing the discovery task itself by focusing on uncovering partial causal structures or specific conditional independencies, as explored by Amini et al. (2022).

In conclusion, we believe that recognizing and addressing the limitations imposed by $\lambda$-strong faithfulness is crucial for the development of more robust and generalizable causal discovery methods. Future research should explore both alternative assumptions and reformulations of the causal discovery task to better align with the complexities of real-world data.

## Impact Statement

This paper presents work whose goal is to advance the field of Machine Learning. There are many potential societal consequences of our work, none which we feel must be specifically highlighted here.

## Acknowledgments

This work was supported by Horizon Europe Programme under GA no. 101120237, project "ELIAS: European Lighthouse of AI for Sustainability" and the National Science Centre (NCN) Grant no. 2020/37/B/ST6/04179. Piotr Miłoś's research was supported by the National Science Center (Poland) grant number 2019/35/O/ST6/03464. We gratefully acknowledge Polish high-performance computing infrastructure PLGrid (HPC Center: ACK Cyfronet AGH) for providing computer facilities and support within computational grant no. PLG/2024/016906.

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

# A. Additional background information

### A.1. $d$-separation

Two nodes $A$ and $B$ in a DAG are said to be $d$-**separated** by a set of nodes $Z$ if all paths between $A$ and $B$ are blocked when conditioning on $Z$. A path is considered blocked under the following conditions:

- If a path includes a non-collider node (a node where arrows do not converge, i.e., a chain or fork), conditioning on that node blocks the path. For example, if $A \to C \to B$, or $A \leftarrow C \to B$, conditioning on $C$ makes $A$ and $B$ independent.

- If the path includes a collider (a node where arrows converge, i.e., $A \to C \leftarrow B$), the path is blocked unless either the collider itself or one of its descendants is conditioned on. For instance, in the path $A \to C \leftarrow B$, conditioning on $C$ or its descendants would unblock the path, making $A$ and $B$ dependent.

- If there are multiple paths connecting $A$ and $B$, all paths must be blocked for $A$ and $B$ to be considered $d$-separated. Even if one path remains unblocked, $A$ and $B$ are d-connected, meaning they are dependent.

In causal discovery, we are interested in making statements about the relationship between the causal graph and the data distribution. Given a causal graph $G$ and the data distribution $P$, the **Markov assumption** states that if variables $A$ and $B$ are $d$-separated in the graph $G$ by some conditioning set $C$, then $A$ and $B$ are conditionally independent in the distribution $P$ when conditioned on the same conditioning set $C$. Formally, this can be written as:

$$A \perp\!\!\!\perp_G B | C \Rightarrow A \perp\!\!\!\perp_P B | C \tag{12}$$

### A.2. Example of faithfulness violation

In this subsection we will illustrate a faithfulness violation for a simple 3 nodes structural causal model with linear functions and additive Gaussian noise. Such a setup is aimed at showing example of faithfulness violation while maintaining simplicity. The example and graphics is from Uhler et al. (2013).

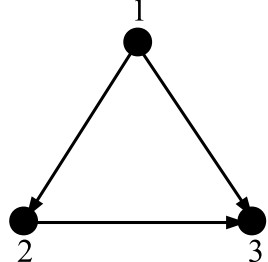

Figure 5: Simple 3 nodes graph $G$.

First lets define a structural causal model on a graph $G$ shown in graph 5.

$$
\begin{aligned}
X_1 &= \varepsilon_1, \\
X_2 &= a_{12} X_1 + \varepsilon_2, \\
X_3 &= a_{13} X_1 + a_{23} X_2 + \varepsilon_3, \\
(\varepsilon_1, \varepsilon_2, \varepsilon_3) &\sim \mathcal{N}(0, I),
\end{aligned}
$$

Since data is linear we can use covariance to measure dependency of variables. Using defined structural causal model, we can write:

$$\text{cov}(X_1, X_2) = a_{12}, \tag{13}$$

$$\text{cov}(X_1, X_3) = a_{13} + a_{12}a_{23}, \tag{14}$$

$$\text{cov}(X_2, X_3) = a_{12}^2 a_{23} + a_{12}a_{13} + a_{23}, \tag{15}$$

$$\text{cov}(X_1, X_2 \mid X_3) = a_{13}a_{23} - a_{12}, \tag{16}$$

$$\text{cov}(X_1, X_3 \mid X_2) = -a_{13}, \tag{17}$$

$$\text{cov}(X_2, X_3 \mid X_1) = -a_{23}. \tag{18}$$

If we define $a_{13}, a_{23}, a_{12}$ in such a way that:

$$a_{13} * a_{23} - a_{1,2} = 0$$

then we get a situation where: nodes 1 and 2 are not d-separated given node 3 in a graph $G$ and $X_1 \perp\!\!\!\perp X_2 | X_3$ which is a violation of faithfulness.

## B. Additional strong faithfulness statistics

**Experimental setup**    In all experiments regarding strong faithfulness statistics, including those in the main paper, we used the following approach. We sampled a 100 graph structures according to each of the graph sampling methods: Erdos-Renyi, small world (Watts-Strogats), scale-free(Barabasi-Albert) and bipartite. Then we used randomly initialized neural networks with 2 layers, hidden dimension 8, and ReLU activation, to generate 1.5M samples from each SCM. Using those datasets we have been able to obtain $\hat{\lambda}$ according to Equation 6.

Most of the graph generation methods return graphs that are guaranteed to be connected. However, Erdos-Renyi graphs can be disjoined if the density parameter is chosen incorrectly. The ER graphs have the property that when the probability of an edge exceeds $\log(n)/n$, where $n$ is the number of nodes, then they are almost surely connected (Erdös & Rényi, 1959). Thus, in case of ER graphs we adjust the density of the graph with its size. We use the following formula for the probability of an edge:

$$p = \epsilon \cdot \frac{\log(n)}{n}, \tag{19}$$

where $\epsilon$ is the density parameter. We consider two scenarios (i) the density is kept the lowest possible to guarantee the graph is connected $\epsilon = 1.1$ (ii) the density is tuned so that we obtain graphs with 10 nodes and expected number of edges 20, which are comparable in terms of density with graphs generated with other methods, $\epsilon = 1.9302$. Table 2 summarizes the densities of the ER graphs. All other types of graphs have expected numbers of edges equal to twice the number of nodes.

| Number of nodes | 4 | 5 | 6 | 7 | 8 | 9 | 10 |
|---|---|---|---|---|---|---|---|
| Expected number of edges $\epsilon = 1.1$ | 2.29 | 3.54 | 4.93 | 6.42 | 8.01 | 9.67 | 11.40 |
| Expected number of edges $\epsilon = 1.9302$ | 4.01 | 6.21 | 8.65 | 11.27 | 14.05 | 16.96 | 20.00 |

Table 2: The densities of ER graphs.

The results are presented in Figures 6 & 7. In all cases we observe rapid vanishing of faithful distributions as graph size or density increases.

## C. NN-opt method details

**Details of experiments with NN-opt method**    In order to test which architecture perform best, we conducted an experiment, training NN-opt method with different sizes of neural networks. The trained models were judged in terms of negative log likelihood and their performance on the task of causal discovery measured as ESHD$_{\text{CPDAG}}$. For each tested architecture, we performed the search for the best regularization coefficient, the tested coefficients were: $[0.1, 0.3, 1.0]$. Among all models, the best results were consistently obtained for regularization coefficient $= 0.3$. The learning rate was set to $0.0003$. The results of the experiments are shown in Table 3. As we can see, the best , both in case of NLL and ESHD$_{\text{CPDAG}}$ was model with two layers and hidden dimension of size 8. Notably this is the same architecture, as was used to generate data.

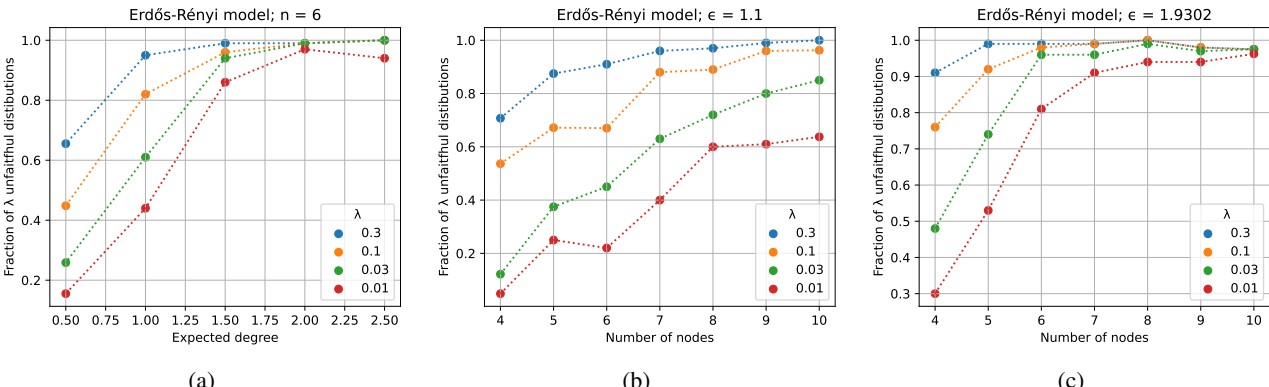

Figure 6: Summary of results of $\lambda$ statistics for Erdős–Rényi graphs.

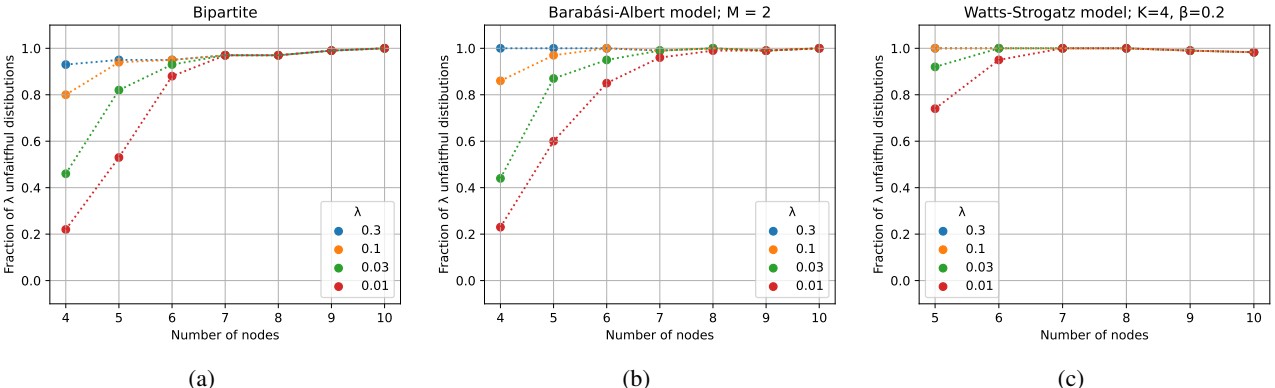

Figure 7: Additional results of $\lambda$ statistics for (a) bipartite graphs, (b) scale-free graphs (genrated using Barabasi-Albert model), and (c) small-world graphs (generated using Watts-Strogatz model). The results are consistent with observations on ER graphs. We observe rapid vanishing of $\lambda$-faithful distibutions.

**Selected hyperparameters:** Number of layers = 2, hidden dimension = 8, regularization coefficient = 0.3.

| Model architecture | NLL | ESHD$_{\text{CPDAG}}$ |
|---|---|---|
| $[4]$ | $0.33_{(0.22,\ 0.43)}$ | $3.63_{(2.83,\ 4.67)}$ |
| $[4, 4]$ | $0.2_{(0.1,\ 0.3)}$ | $3.15_{(2.0,\ 4.65)}$ |
| $[4, 4, 4]$ | $0.23_{(0.14,\ 0.34)}$ | $3.03_{(2.33,\ 4.07)}$ |
| $[8]$ | $0.18_{(0.06,\ 0.29)}$ | $2.13_{(1.43,\ 3.07)}$ |
| $[8, 8]$ | $\mathbf{0.13}_{(0.02,\ 0.24)}$ | $\mathbf{1.23}_{(0.77,\ 1.87)}$ |
| $[8, 8, 8]$ | $0.22_{(0.12,\ 0.32)}$ | $2.77_{(1.97,\ 3.67)}$ |
| $[16]$ | $0.14_{(0.03,\ 0.26)}$ | $1.77_{(1.1,\ 2.73)}$ |
| $[16, 16]$ | $0.33_{(0.24,\ 0.42)}$ | $2.4_{(1.0,\ 4.32)}$ |
| $[16, 16, 16]$ | $0.88_{(0.8,\ 1.0)}$ | $4.0_{(3.07,\ 4.97)}$ |

Table 3: The performance of NN-opt method models with different architectures. The numbers in the subscripts, correspond to 0.95 confidence intervals. The experiments were performed on 30 graphs.

## D. Errors in $\hat{\lambda}$ computation

We evaluate the quality of $\hat{\lambda}$ estimation. In Table 4 we report the bootstrap standard error and bootstrap bias for estimates $\hat{\lambda}$. We use a standard data generation protocol based on ER graphs and randomly initialized NNs, as described in the main text. We generate datasets for 10 random SCMs and report aggregate statistics for two sample sizes: 1.5M as was used to generate plots in Section 3.1 and Appendix B and 100K as was used in Section 4. The bootstrap size was 100 due to computational constraints.

| | | Min | Max | Mean | Median |
|---|---|---|---|---|---|
| 100K samples | Bootstrap Standard Error | 2.00e-3 | 9.38e-3 | 3.93e-3 | 3.19e-3 |
| | Bootstrap Bias | 4.73e-5 | 5.92e-3 | 2.84e-3 | 3.92e-3 |
| 1.5M samples | Bootstrap Standard Error | 6.17e-4 | 1.15e-2 | 2.2e-3 | 7.98e-4 |
| | Bootstrap Bias | 5.25e-6 | 1.19e-2 | 1.71e-3 | 1.81e-4 |

Table 4: Aggregated statistics of bootstrap standard error and bias of $\hat{\lambda}$ estimation.

## E. Details About Benchmark and Extensions

### E.1. Dataset generation details

The data is generated using a fully connected MLP with two hidden layers of 8 units each, initialized with random weights drawn from a standard normal distribution and use the ReLU (Nair & Hinton, 2010) activation function to introduce non-linearity. The neural network models the relationships between variables in the underlying DAG, where each node represents a variable and the edges capture dependencies between these variables. The input variables, which serve as the initial causes in the graph, are sampled from normal distributions. The noise added to the system is sampled from a Gaussian distribution $\mathcal{N}(0, 0.1^2)$, simulating uncertainty in the model. The dataset consists of 100,000 data points, and the data is rescaled to maintain consistency across samples.

### E.2. Model hyperparameters

We performed extensive hyperparameter tuning for all methods. In addition to the MLP architecture grids described in Appendix E.4, the following hyperparameter grids were explored:

**DCDI   Grid search**: Regularization coefficients tested: [0.1, 0.3, 1, 2]. Values below 0.001 or above 5 led to poor performance. **Selected**: Regularization coefficient = 1, learning rate = 0.001, Augmented Lagrangian tolerance = $10^{-8}$.

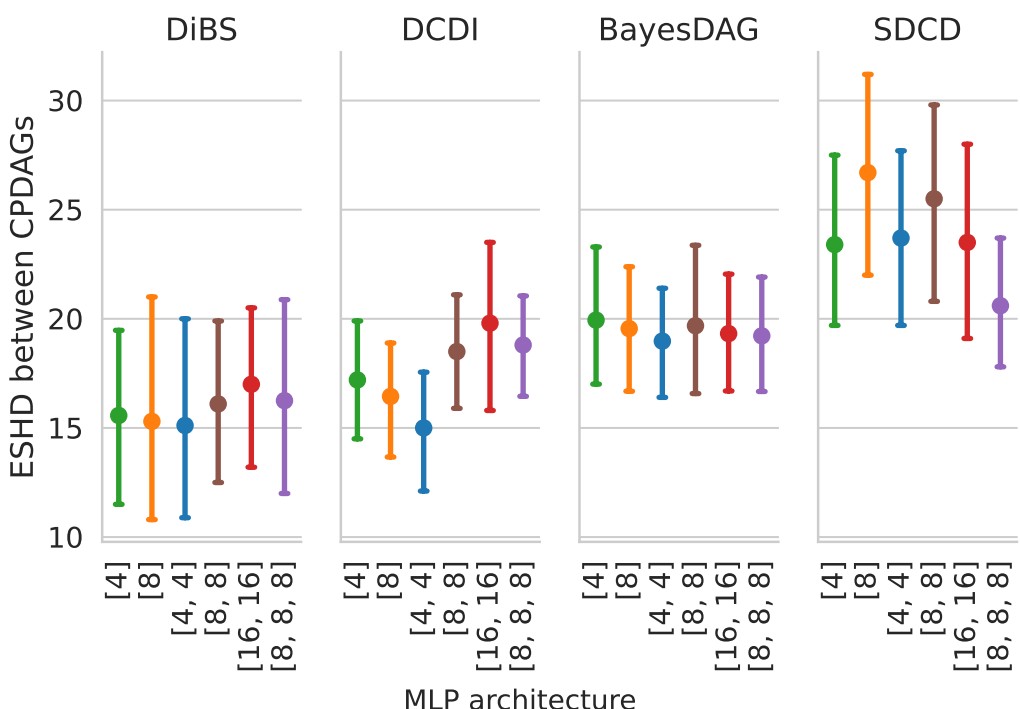

Figure 8: Comparison of ESHD$_{\text{CPDAG}}$ using different MLP architectures as functional approximator for ER(10, 2) dataset and 800 observational samples, averaged over 30 samples.

**DiBS**   **Grid search**: Alpha linear: [0.01, 0.02, 0.05], kernel parameters: h latent: [0.5, 1.0, 2.0], h theta: [20.0, 50.0, 200.0], step size:[0.05, 0.03, 0.01, 0.005, 0.003]. **Selected**: Alpha linear = 0.02, h latent = 1.0, h theta = 50.0, step size = 0.03.

### E.3. Testing models architecture

Finally, we investigate the impact of the neural model architecture, used as the functional approximator, on the performance of the benchmarked methods. Specifically, we assess how the capacity of different architectures influences the ability to uncover causal relationships from synthetic data. To provide a comprehensive evaluation, we explored architectures with 1, 2, and 3 layers, configured with 4, 8, and 16 hidden units.

Results, presented in Figure 8 show the comparison of ESHD$_{\text{CPDAG}}$ metric for the benchmarked architectures across all methods on dataset with 800 samples. We find that the choice of neural architecture has no significant impact on performance across methods. We conclude that any of the tested MLP architectures provides sufficient capacity to model the underlying distribution effectively. Additionally for BayesDAG and SDCD we implemented layer normalization and residual connections. We investigated the impact of this changes in architectures and did not found any significant differences, see Figure 13. The details and additional experimental results are in Appendix E.4.

**BayesDAG**   **Grid search**: Scale noise: [0.1, 0.01], scale noise p: [0.1, 0.01, 1.0], lambda sparse: [50.0, 100.0, 300.0, 500.0]. **Selected**: Scale noise = 0.1, scale noise p = 0.01, lambda sparse = 500.0.

**SDCD**   **Grid search**: Constraint modes: ["exp", "spectral radius", "matrix power"]. The ESHD$_{\text{CPDAG}}$ metric showed similar results across modes. **Selected**: Spectral radius was chosen for faster computation, with a learning rate of 0.0003.

For each of these method, all other parameters were retained from the original paper or code.

### E.4. Model architecture comparision within method

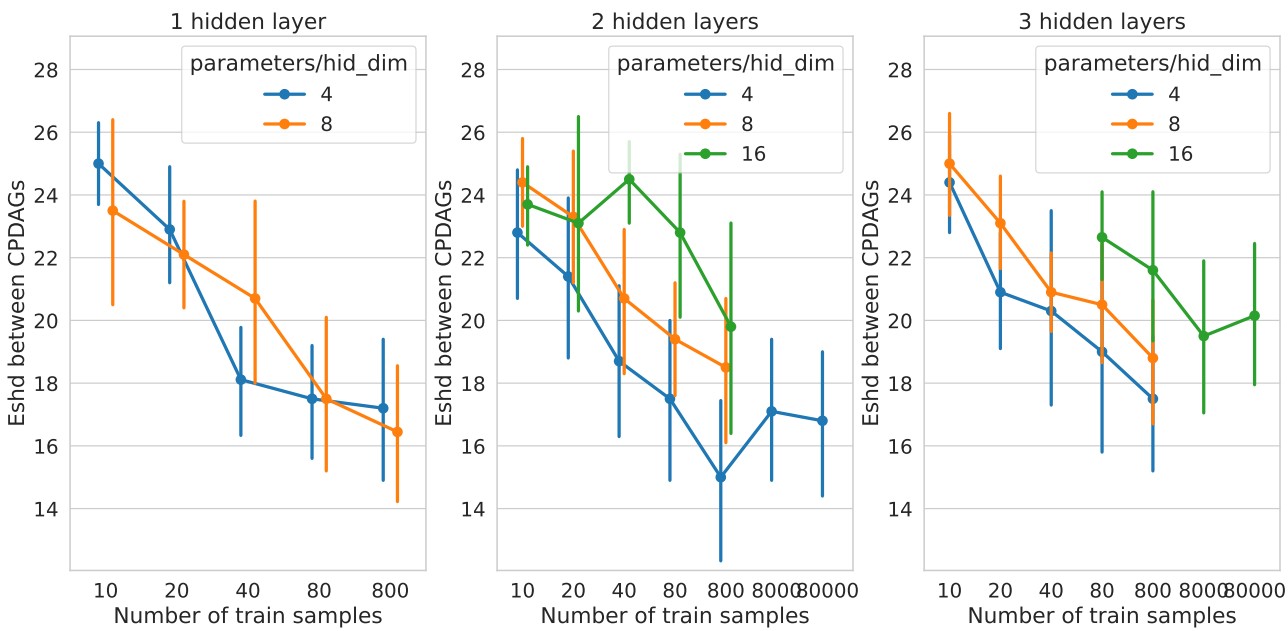

Figure 9: Comparison of the ESHD_{CPDAG} of DCDI for datasets with different observational sample size. The result is based on 10 graphs.

**DCDI**    In Figure 9, we present the performance analysis of the DCDI across various neural network configurations. Our results reveal that the optimal performance is generally achieved by a two-layer model with a hidden dimension of 4. Interestingly, we observe that more expressive models exhibit diminished performance relative to the smaller models.

**DiBS**    Figure 10 presents the performance analysis of the DiBS method across various neural network configurations. As with the DCDI method, we evaluate models with different numbers of layers and hidden dimension sizes. Consistent with DCDI, we find that the optimal performance for DiBS is achieved by a two-layer model with a hidden dimension of 4. However, the performance landscape for DiBS exhibits less variability across different model configurations. Single-layer models perform nearly as well as the optimal two-layer model.

Furthermore, we observe that more expressive models do not show a significant degradation in performance as was seen with DCDI. The overall differences in metric across all tested configurations are relatively small for DiBS, indicating a more consistent performance across varying levels of model complexity.

**BayesDAG**    Figure 11 compares the performance of BayesDAG across different model architectures and sample sizes. For smaller sample sizes, BayesDAG's performance remains consistent, with noticeable differences emerging only at a sample size of 800. This suggests that BayesDAG requires more data to fully leverage its model capacity, unlike what we observed for DCDI and DiBS, where performance varied more significantly across sample sizes. Notably, the best-performing architecture for DiBS is a two-layer MLP with a hidden dimension of 4.

**SDCD**    Figure 12 presents a similar comparison of SDCD performance across different MLP architectures and sample sizes. Interestingly, the three-layer architectures show stagnant performance regardless of sample size, while the one-layer models exhibit significant improvement as the sample size increases. Overall, the best performance is achieved with a one-layer MLP with 8 hidden units, although it remains comparable to the one-layer MLP with 4 hidden units and the two-layer MLP with 4 hidden units.

**Model architecture**    Inspired by BayesDAG, we also implemented layer normalization and residual connections to assess their impact. We conducted additional experiments on both the best-performing model ([4, 4]) and the largest model ([8, 8, 8]). The size of networks was similar to the one proposed in articles introducing tested methods: in DCDI it was [16, 16], for SDCD it was [10, 10], for DiBS [5, 5] and for BayesDAG it was a two layer network with a hidden size varying

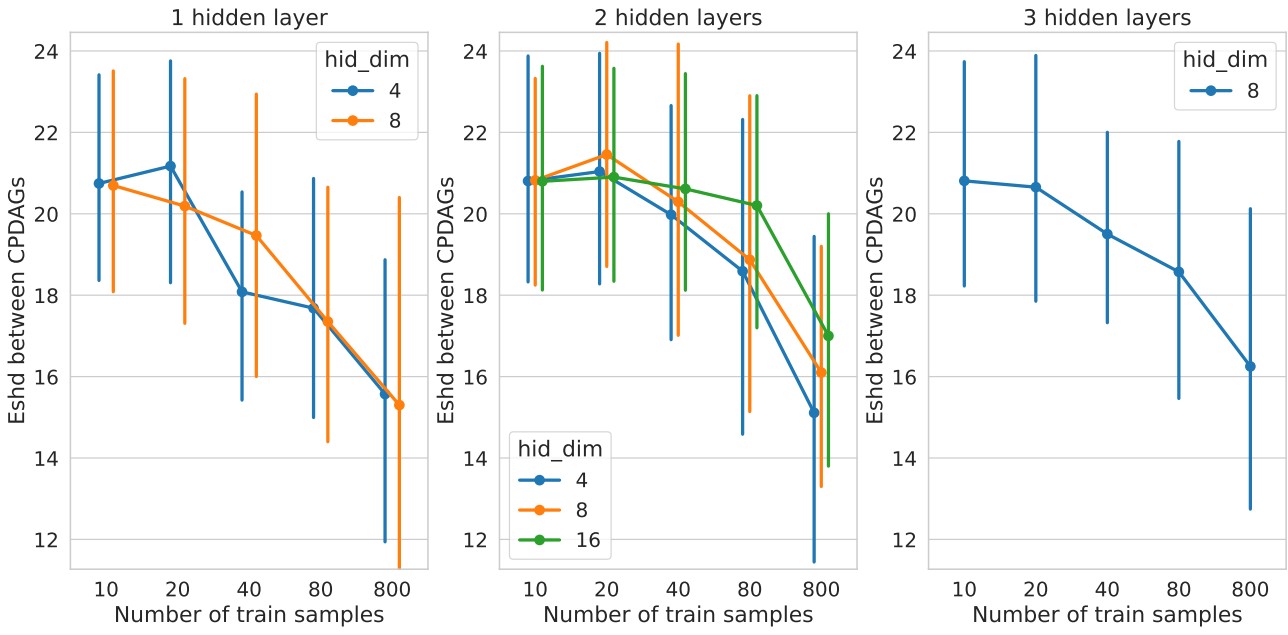

Figure 10: Comparison of the performance of DiBS depending on the model architecture and number of samples.

with dimensionality. The results of these tests are presented Figure 13. We show, there is no significant and consistent improvement across all networks, supporting our initial conclusion that variations in MLP architecture have minimal impact on performance.

## F. Justification of evaluation metrics

We design metrics based on popular SHD, F1-score metrics, which we explain shortly below.

**The Structural Hamming Distance.** SHD (Tsamardinos et al., 2006) quantifies the difference between the predicted graph and the ground truth graph by counting the number of edge insertions, deletions, and reversals required to transform one into the other. SHD values indicate the degree of error in recovering the true causal structure: lower SHD values signify better predictions, while higher values indicate more significant discrepancies.

**The F1-score.** The F1-Score measures the harmonic mean of precision and recall for edge predictions, where precision reflects the fraction of correctly predicted edges among all predicted edges, and recall reflects the fraction of correctly predicted edges among the true edges.

We evaluate causal discovery methods based on observational data. In general, in this setup, it is only possible to recover true DAG up to a Markov Equivalence Class, a class of graphs with the same conditional independence relationships, due to identifiability issues TODO cite pearl?. If we were to compare the predicted and ground true graphs using standard metrics like SHD or F1-score we would obtain distorted results — graphs from the MEC class do not generally receive these metrics' optimal values.

Therefore, we modify the formulation of the metrics to account for the limitations of causal discovery from observational data. We define ESHD$_{\text{CPDAG}}$ and F1-Score$_{\text{CPDAG}}$. These metrics attain their optimal values, 0 and 1 correspondingly, for all DAG from ground truth MEC. Additionally, some of the benchmarked methods are Bayesian thus return the posterior over possible solutions. For those methods, we design metrics that compute the expected value over the posterior and approximate it with the Montecarlo estimator based on a sample of size 100.

We define **Expected SHD between CPDAGs** as:

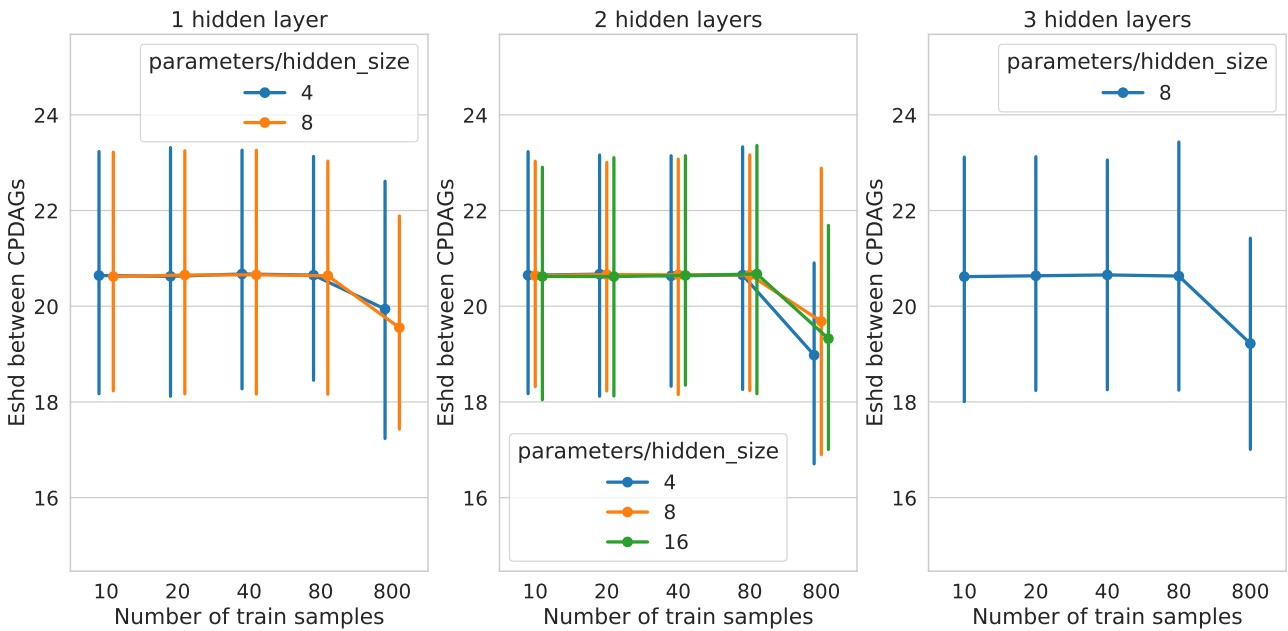

Figure 11: Comparison of the performance of DiBS depending on the model architecture and number of samples.

$$\text{ESHD}_{\text{CPDAG}}(\mathcal{G}, \mathbb{G}) = \mathbb{E}_{\mathcal{G}^* \sim \mathbb{G}}[\text{SHD}(\text{CPDAG}(\mathcal{G}), \text{CPDAG}(\mathcal{G}^*))], \tag{20}$$

where $\mathbb{G}$ is the resulting distribution of graphs, $\mathcal{G}^*$ is a graph sampled from $\mathbb{G}$ and $\mathcal{G}$ is the ground true graph. Similarly, we compute the **Expected F1-Score between the CPDAGs**:

$$\text{F1-Score}_{\text{CPDAG}}(\mathcal{G}, \mathbb{G}) = \mathbb{E}_{\mathcal{G}^* \sim \mathbb{G}}[\text{F1-Score}(\text{CPDAG}(\mathcal{G}), \text{CPDAG}(\mathcal{G}^*))]. \tag{21}$$

subsectionInfluence of sample samples on performance on the graph with ER(5, 1)

Figure 14 shows the $\text{ESHD}_{\text{CPDAG}}$ of benchmachmarked methods for different sample sizes. For all observational sample sizes, SDCD and DCDI have a large confidence interval. For datasets with 2,500 and 8,000 samples, BayesDAG performs better than other benchmarked methods, getting small confidence interval for 8,000 samples.

## G. Algoritm pseudocode

Pseudocode of the method described in Sec. 3 is provided in Algorithm 1.

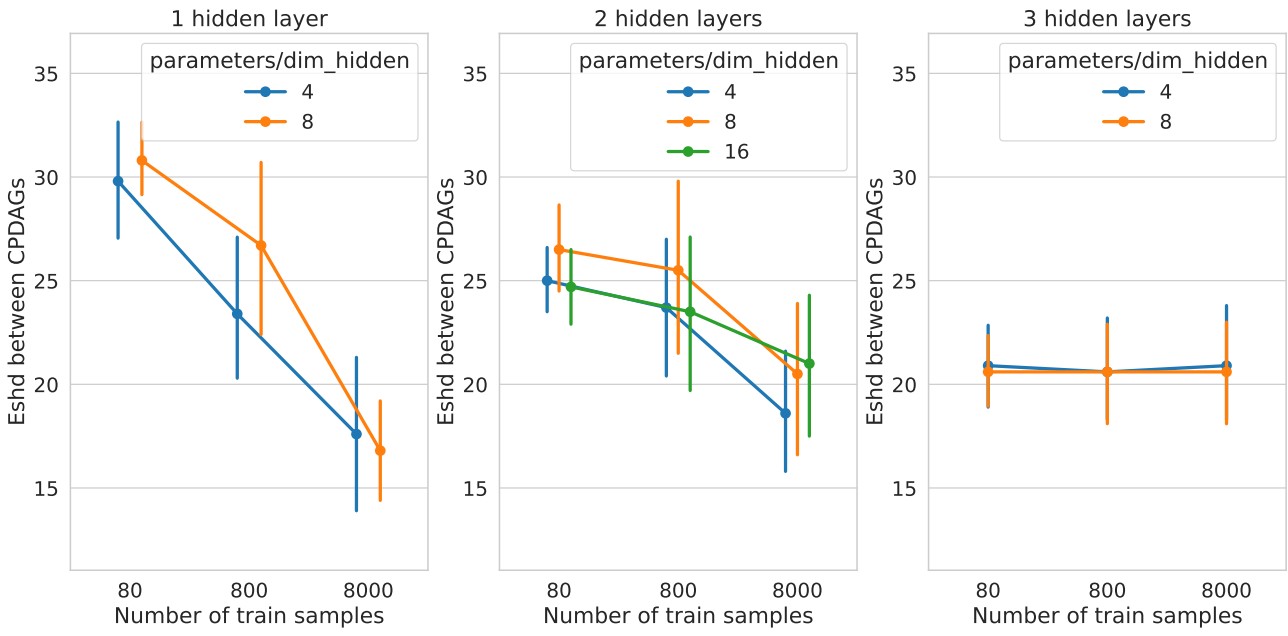

Figure 12: Comparison of the performance of SDCD depending on the model architecture and number of samples.

---

**Algorithm 1** Overview of NN-Opt

---

1: **Input:** Set of nodes $V$, training data $\{D_i\}_{i \in V}$ , regularization coefficient $\lambda$, $\mathbb{G}$ the space of DAGs with nodes $V$
2: *# Part 1: Network fitting*
3: **for** $i \in V$ and $\pi \subseteq V \setminus \{i\}$ **do**     ▷ For each variable and each possible parent set
4:     $\theta_{i,\pi} \leftarrow \text{TRAINNETWORK}(i, D, \pi)$     ▷ Train ensembles of 3 networks
5: **end for**
6: *# Part 2: Exhaustive graph search*
7: **for** $G \in \mathbb{G}$ **do**     ▷ Evaluate all possible DAGs
8:     $\text{score}_G \leftarrow \sum_{i \in V} \text{COMPUTENLL}(D_i, D_{Pa_i^G}, \theta_{i,Pa_i^G})$     ▷ Compute NLL using ensemble
9:     $\text{score}_G \leftarrow \text{score}_G + \lambda \cdot |G|$     ▷ Add regularizing term
10: **end for**
11: **Output:** $\arg\max\{\text{score}_G : G \in \mathbb{G}\}$

---

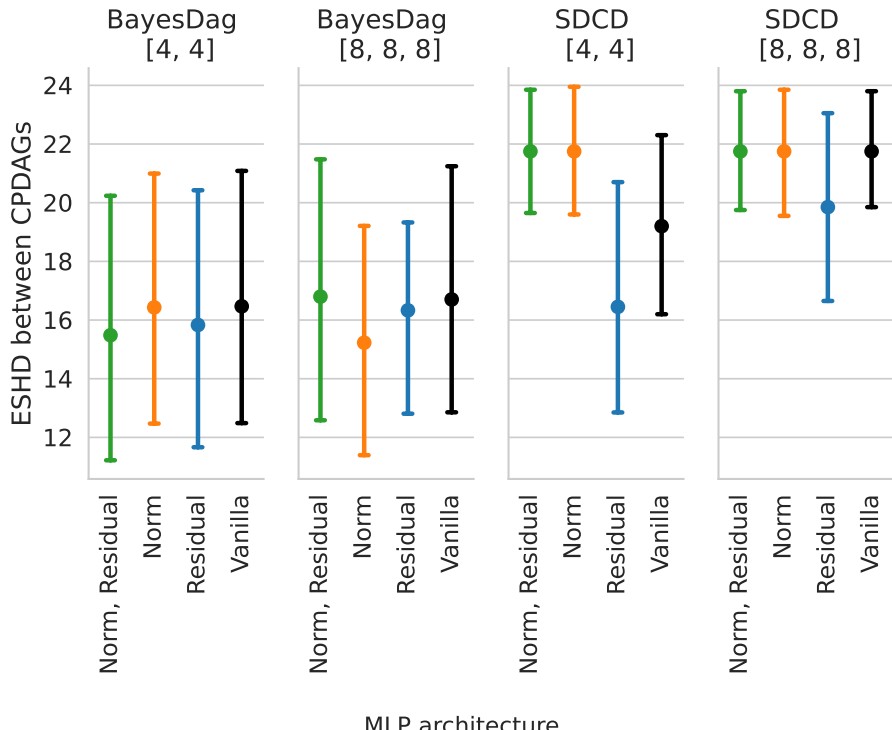

Figure 13: Comparison of the performance of SDCD depending on the model architecture and number of samples.

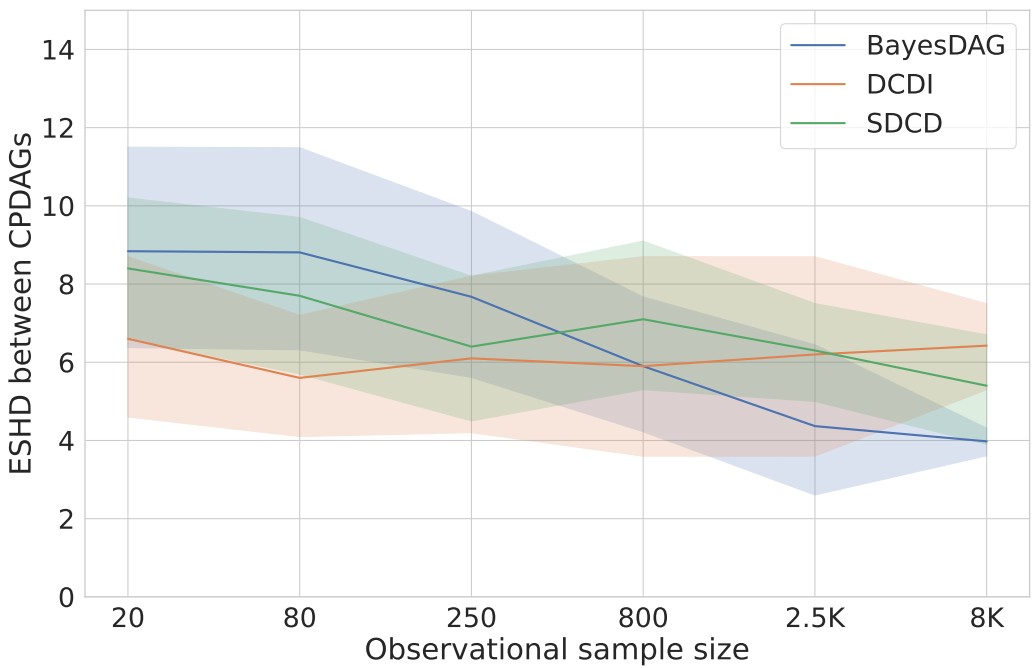

Figure 14: Comparision of $ESHD_{CPDAG}$ for benchmarked methods on ER(5, 1) dataset, averaged over 10 graphs.

