# OpenReview forum: "Since Faithfulness Fails: The Performance Limits of Neural Causal Discovery"
_ICML.cc/2025/Conference — ICML 2025 poster_

### Official Review · Reviewer_ERvU · 2025-03-14

**Overall Recommendation:** 4

**Summary:**

The paper benchmarks several representative neural causal discovery methods in a coherent and charitable way, revealing consistent shortcomings. These are attributed to faithfulness violations, even in large sample sizes and small graphs, suggesting a more fundamental flaw in the neural causal discovery paradigm.

**Claims And Evidence:**

I generally found the claims clearly stated and well-supported, with two important exceptions:
1. it's claimed that neural methods can't detect absence/existence of causal relationships, but it's not clear to me that randomly initialized NNs are guaranteed to have 'strong enough' influence; in linear simulations, it's common to sample weights $\mathrm{abs}(w) > 0.5$, but it's not if clear anything similar was done here
2. NNs are universal function approximators, so it's unclear to me how a different approximator is going to solve the problem; rather seems to me different objective functions are needed---and the paper doesn't give evidence that it's specifically a NN problem, disentangled from the similar objective functions the neural approaches are using.

And some less important exceptions:
3. In Section 5: "These parameter choices align with commonly studied medium-sized graphs in causal discovery research". I wouldn't say 5- and 10-node graphs are medium-sized, and expected degrees of 1 and 2 are quite sparse.

**Essential References Not Discussed:**

Nothing missing that I'm aware of.

**Experimental Designs Or Analyses:**

Yes, these all seem reasonable, other than related to points 1 and 3 in _Claims and evidence_ above.

**Methods And Evaluation Criteria:**

Yes, these all seem reasonable.

**Other Comments Or Suggestions:**

- second paragraph in intro: should be "ground-truth" instead of "ground-though"
- end of paragraph after (5): should be "bridge the gap" instead of "breach..."
- in Section 3: should be "Synthetic" not "Synthetics"
- missing label in caption of Figure 1(b), so it appears as ??
- __conclusion__ before Section 4: should be "that" instead of "hat"
- Section 4.2: (twice) should be "faithful" instead of "faithfull"
- Line 747: "TODO cite pearl?."

**Other Strengths And Weaknesses:**

Already covered.

**Questions For Authors:**

1. Have the authors tried using linear or NN simulations with stronger relations, like mentioned in __Claims and evidence__ above?
2. Likewise, do the authors have an argument against neural approaches specifically or for any other particular approaches, independent of the objective functions used?
3. Have the authors tried comparing non-neural methods? Do they fare any better?
4. In __Conclusion__ before Section 4, what does "the number of equivalent graphs will reach 0" mean?

Satisfactory answers to the first 3 questions would improve my overall recommendation.

**Relation To Broader Scientific Literature:**

The findings here more systematically and rigorously support and relate previous findings concerning $\lambda$-faithfulness and poor performance of neural causal discovery methods.

The paper doesn't really make claims about non-neural causal discovery, but it would interesting to see at least some standard/state of the art non-neural methods included for comparison, such as kernel PC or GRaSP.

**Theoretical Claims:**

N/a

---

> ### Author Rebuttal · Authors · 2025-04-01
>
> We sincerely appreciate the Reviewer's positive feedback and thoughtful assessment of our work. We are glad that our benchmarking of representative neural causal discovery methods was recognized as both coherent and charitable, providing a clear and systematic evaluation of their limitations. We are also pleased that our claims were found to be well-supported. Moreover, we are grateful for the acknowledgment that our findings rigorously build upon and relate to prior work.
>
> To clarify our claims (addressing feedback from *Claims And Evidence* section):
>
>
> 1. We claim that neural networks cannot reliably detect the absence or existence of causal relationships given practically available data volumes. The strength of causal relationships plays a crucial role; we quantify it using the notion of λ-faithfulness. We show that the required sample size will be infeasible in practical scenarios.
>
>     We thank the Reviewer for suggesting the additional experiment with artificially ‘strong influence’. For the linear case, such an analysis was performed in [Uhler2013] (see Figure 5 in their paper). They show that increasing the strength does not change the overall picture. This might seem counterintuitive at first. However, increasing the direct links does not eliminate the exponential vanishing of λ-faithful distributions due to the cancellation of paths.
>
>     We highlight this information in the revised version of the paper. Moreover, we run an analogous simple experiment for the non-linear case. Namely, during data generation, neural network weights were initialized with values spaced from zero by a factor c∈{0.0,0.5,1.0}. Importantly (and akin to the linear case), this did not result in observable differences in the distribution of the λ-property, see Figure R.2 in the [rebuttal material](https://drive.google.com/file/d/1m7rFfvf2_xoQorCprk6RE1MiITmYivhy/view?usp=sharing)
>
> 2. The reviewer is completely right. Using different approximators would unlikely solve the problem, and indeed, novel score functions or an alternative causal discovery objective are needed. We are sorry for the confusion. We have improved the description so that it is now explicitly stated.
> During the rebuttal, we provide another piece of evidence by using kernel-based PC (i.e., a non-NN approximator), observing that the slow convergence phenomenon is present. We include this result in Figure R.3 in the [rebuttal material](https://drive.google.com/file/d/1m7rFfvf2_xoQorCprk6RE1MiITmYivhy/view?usp=sharing)
>
> 3. We understand the reviewer's concerns regarding the size of the evaluation graphs. We will revise the description to refer to “small and medium-sized graphs”. (We also note that in Section 5, we use graphs with 30 nodes.)
>
>
> As for the questions:
> 1. Please see our response to Claims and Evidence 1.
> 2. Please see our response to Claims and Evidence 2.
> 3. During the rebuttal, we compared kernel-based PC to our algorithm introduced in Section 3. As expected, we observed comparable performance.
> 4. The referenced sentence pertains to the experiment in Section 3.1. Our intended meaning is that the number of structures outside the MEC class that receive statistically equivalent scores decreases slowly as the number of samples increases. We will clarify this in the revised version of the paper.
>
> Again, we thank the reviewer for the constructive criticism. The new version of the manuscript includes the textual improvements announced above and several other small amendments. Moreover, it includes the new experimental results. We’d be happy to address any additional questions or concerns.
>
> [Uhler2013] Uhler, Caroline, et al. "Geometry of the faithfulness assumption in causal inference." The Annals of Statistics (2013): 436-463

---

> > ### Comment · Reviewer_ERvU · 2025-04-03
> >
> > The rebuttal addresses most of my concerns, so I increase my overall recommendation from 3 to 4.
> >
> > As a final comment, I suggest the authors try to phrase some of the claims of the paper more carefully, paying special attention to whether each claim/evidence concerns neural networks, (penalized likelihood-based) score functions, or the intersection of the two. For example, do the conclusions drawn from Figure 1(a) hold across all values of $\gamma$? And how does it compare to using the MLE in the linear gaussian (or other parametric) setting? I wonder if some of the claims in the paper should actually be about penalized likelihood-based scores (which many NN methods use), rather than NNs themselves. Another interesting comparison in this vein would be to see if differentiable ANM methods (which use a different score, but still use NNs) also succumb to $\lambda$-unfaithful samples.
> >
> > Overall, I think the paper is very thought-provoking and contributes a valuable, critical perspective among the growing number of neuro-causal methods.

---

> > > ### Author Response · Authors · 2025-04-07
> > >
> > > We thank the reviewer for the positive feedback, the increased score, and the kind comments regarding the significance and impact of our work. We also appreciate the suggestion to clarify the claims, which we will carefully consider as we prepare the next version of the paper.
> > >
> > > We thank the reviewer for suggesting extending the analysis. Exploring other MLE-based methods and recent differentiable ANM approaches is indeed an interesting and important direction for future work, and we expect that our results will continue to hold in these settings.

---

### Official Review · Reviewer_JYG6 · 2025-03-14

**Overall Recommendation:** 2

**Summary:**

This paper claims that while neural causal discovery methods have become more scalable, they fundamentally struggle with accuracy when identifying true causal relationships. Neural networks are unable to reliably distinguish between real and non-existent causal connections in finite samples, and violations of the faithfulness property—which occur frequently in practice—significantly undermine their performance. The authors conclude that these limitations require a paradigm shift in the field of neural causal discovery.

**Claims And Evidence:**

I feel that I don't understand section 3. It seems like section 3 shows that as sample size increases, the neural network does better and better at identifying the structure. Then the conclusion claims that methods can't identify structure consistently. The prose directly above that conclusion states that larger samples enables identification of structures in more difficult datasets. How can I reconcile these two things?

The metrics of lamba and lambda hat make sense to me, and its cool that these measures correlate with the number of samples needed for convergence.

**Essential References Not Discussed:**

N/A

**Experimental Designs Or Analyses:**

The experimental design of using neural networks to generate a dataset and then training neural networks to perform causal discovery on that dataset seems reasonable to me if the goal is to problematize their ability to do causal discovery.

**Methods And Evaluation Criteria:**

I am not familiar with the state of the art methods in neural causal discovery, so I can't say whether the methods they chose are an appropriate spread of the state-of-the-art models.

**Other Comments Or Suggestions:**

245 "hat the number" -> "that the number"

Figure 1 caption has Section ??

line 290 Faithfull

**Other Strengths And Weaknesses:**

I feel like I might be missing the main point of this paper. Section 5 has prose that references section 3 as if section 3 showed that neural network models don't scale with the number of examples provided in training, but that is not what section 3 says at all. It says that neural models get better the more data that is provided.

Could you explain how Figure 1b and Figure 3b are saying the same thing? From my perspective, they are showing very different results.

Separately, if the whole point of this paper is arguing for a paradigm shift in causal discovery away from neural networks, how can this whole paper only use artificial datasets? I don't think a field should progress by people constructing difficult datasets that show methods fail, shouldn't we be grounded out in some sort of real world dataset or phenomena?

"It may hold, though is highly unprobable, that real-world distributions adhere to λ-strong faithfulness despite large sizes of the graph. Further investigation is required"
This quote from the end of section 6 is really important to me. I think making a technical point about NCD methods and where they struggle is totally a reasonable and good thing to do. However, if you want to argue that the field needs a paradigm shift, I feel you must ground out this claim in an evaluation of the real world data the field someday wants to model. Without an argument that the assumptions you made in the paper will hold in real-world setting, I feel I have to reject this paper based on how ambitious and far-reaching the introduction, title, and abstract are.

**Questions For Authors:**

N/A

**Relation To Broader Scientific Literature:**

I think understanding how to leverage neural networks for causal discovery is broadly of huge importance, and showing their limitations can be very helpful in that process.

**Theoretical Claims:**

N/A

---

> ### Author Rebuttal · Authors · 2025-04-01
>
> We appreciate the Reviewer's thoughtful comments and acknowledgment of our work's importance in understanding neural causal discovery limitations. We admit shortcomings in the presentation, including those pointed out by the Reviewer. We have made a substantial effort to improve the quality.
>
> Below, we clarify the specific concerns. To be on the safe side, we start with a short general summary.
>
> Causal graph discovery has traditionally been framed as a discrete combinatorial problem. More recently, neural network-based continuous optimization methods have been introduced, offering scalability and computational efficiency. Our study reveals the key limitations of such approaches: likelihood-based methods that utilize neural networks struggle to distinguish between true and spurious causal connections when using realistically available data volumes. We show that the problem quickly exacerbates with the graph size.
> From this, we draw our key takeaway: the casual discovery community needs to search for alternative objective functions.
>
>
> With this in mind, we now address the Reviewer's specific concerns.
>
>
> **Regarding results from sec. 3.** The Reviewer points out a discrepancy between the improvement reported in Section 3 and our conclusion about the fundamental limitations. The latter stems from the observation, which is key to the whole paper, that even using substantial amounts of data (e.g., 8,000 samples), our idealized method fails to identify even small causal structures (5 nodes). Subsequently, we have confirmed that the problem persists even with 80,000 samples (see Figure R.3 in the [rebuttal material](https://drive.google.com/file/d/1m7rFfvf2_xoQorCprk6RE1MiITmYivhy/view?usp=sharing))
>
> Moreover, the results shown in  Sec. 4 discuss how distributions associated with larger graphs quickly become highly complex, leading to a sharp increase in data requirements. Together, these results show that the structure identification of large graphs requires unrealistically large datasets.
>
> On the conceptual level, the difficulty of discovery (i.e., the number of required samples) is highly correlated with $\lambda^{-1}$ from the $\lambda$-faithfulness notion. In Section 4, we show that $\lambda$ is typically small and decreases with the size of the graphs.
>
> We will provide an improved conclusion paragraph that clearly states the above in the next revision of the paper.
>
> **Regarding sections 3 and 5 showing the same thing.** Both graphs convey the same message: that casual discovery is not achievable using a practically available data regime. Sec 3 & 4 (Fig. 1b) show that large graphs require impractically large datasets (as discussed above), even when using idealized methods. Sec 5 (Fig. 3b) reinforces these findings, specifically, when using a practical method, we observe slow (or lack of) convergence.
>
> We hope this clarifies the issue and will revise Sec. 5 accordingly. Please let us know if further questions arise.
>
> **Regarding the usage of synthetic datasets.** We acknowledge concerns about our reliance on synthetic datasets, a necessity due to the absence of large, real-world datasets annotated with ground-truth causal graphs. However, we follow standard causal discovery practices, using widely accepted synthetic benchmarks that are randomly generated rather than adversarially designed, as in prior works. Thus, we believe the results are generalizable to real-world datasets.
>
> In our rebuttal, we strengthen our results by providing lambda statistics for additional graph structures (scale-free, small-world, and bipartite), ensuring broader coverage of realistic scenarios, see Figure R.1 in the [rebuttal material](https://drive.google.com/file/d/1m7rFfvf2_xoQorCprk6RE1MiITmYivhy/view?usp=sharing). Given these points, we believe our methodology is well-justified, but we are open to discussing further refinements if the Reviewer has specific suggestions.
>
> ***
>
>
> We sincerely thank the Reviewer for their careful reading and for identifying the typographical errors. We will correct these and incorporate the requested neural network details in the final version.
>
> We hope that this rebuttal clarifies the Reviewer's concerns and highlights the significance of our findings. We believe our work provides valuable insights into the limitations of current neural causal discovery methods and motivates the need for alternative objective functions. In light of these clarifications, we respectfully ask the Reviewer to reconsider their recommendation.
>
> While we have endeavored to address all questions thoroughly, the character limit required us to keep our responses concise. We would be happy to provide further explanations if needed.

---

> > ### Comment · Reviewer_JYG6 · 2025-04-02
> >
> > I won't be changing my score, as I don't really understand how 8k or 80k are impractically large numbers when it comes to datasets. I also don't really understand how you can claim results are generalizable to real world datasets.
> >
> > That being said, I recognize I may be missing some of the main points, and if the other reviewers and area chair agree this paper should be accepted, I'm perfectly happy with that!

---

> > > ### Author Response · Authors · 2025-04-04
> > >
> > > We appreciate the reviewer's feedback and positive outlook on our work. We make an effort to address Reviewers' concerns around the claims of the paper.
> > >
> > > **Regarding data size**
> > >
> > > Collecting data from real-world causal systems is usually a costly, time-consuming process, for example, including wet lab experiments for the use cases in biology or chemistry. Thus, the community standard is to focus on datasets that might feel small compared to the standard of other ML fields, even in the cases when synthetic data are used. For reference, we present a summary of datasets used in causal discovery for context, using N for nodes and S for samples.
> > >
> > > [DCDI]:
> > > - Sachs: N=11, S~=6000
> > > - synthetic: N = {10, 20}, S=10K
> > >
> > > [BayesDag]:
> > > - synthetic: N=5, S=500; N={30, 50, 70, 100}, S=5K
> > > - Syntren (semi-synthetic, simulation): N=20, S=500
> > > - Sachs: S=800 (only observational), N=11
> > >
> > > [GraN-Dag]
> > > - Appendix A.4, titled “large sample size experiment”: N50, S<=10K
> > > - synthetic: N={10, 20, 50, 100}, S=1K
> > >
> > > [CAM]
> > > - Real data: N=39, S=118
> > > - synthetic: N={10, 100, 1000}, S=200
> > >
> > > [SDCD]:
> > > - synthetic: N={20, 30, 40}, S=10K
> > >
> > > [DiscoGEN]:
> > > - synthetic: N=100, S<=50K
> > >
> > > [AVICI]:
> > > - training: N=(2,50), S=200,
> > > - evaluation: N=(2,50), S={30, 100, 300, 1000}
> > >
> > > [FIP]:
> > > - training: N=100, S=200
> > > - evaluation; N=100, S<= 10K
> > >
> > > When it comes to real-world data, we would like to refer to Table 2 from [Grounding] describing the biological data:
> > >
> > > | Dataset | Description | Number of interventions | Number of samples | Number of nodes |
> > > |---------|-------------|------|-------|------|
> > > | Wille et al. (2004) | Gene expression microarray (*A. thaliana*) | 1 | 118 | 39 |
> > > | Dixit et al. (2016) | Perturb-seq (bone marrow-derived dendritic cells) | 8 | 14427 | 24 |
> > > | Replogle et al. (2022) | Perturb-seq (cell line K562) | 1158 | 310385 | 8552 |
> > > | Replogle et al. (2022) | Perturb-seq (cell line RPE1) | 651 | 247914 | 8833 |
> > > | Frangieh et al. (2021) | Perturb-CITE-seq (melanoma cells) | 249 | 218331 | 1000 |
> > > | Sachs et al. (2005) | Flow cytometry (CD4+ T cells) | 6 | 5846 | 11 |
> > >
> > >
> > > Even though the largest dataset contains ~310k samples, we emphasize that the ratio of dataset size to graph size is significantly worse in their case compared to ours (we use 80k samples with a graph size of 5, whereas their graph size is 8k). Moreover, the difficulty of the causal discovery problem increases rapidly with graph size, at least when measured using the proxy of lambda, as indicated in Section 4. Given these factors, we strongly believe our results support the conclusion that 'causal discovery is impossible in the listed cases under the current paradigm’.
> > >
> > > At the same time, we have partial results for a dataset of 800k samples. Although these results lack statistical power, they are fully consistent with our claims. We commit to expanding this part in the camera-ready version.
> > >
> > > **Regarding real-world datasets**
> > >
> > > We acknowledge that real-world datasets often exhibit complexities that synthetic datasets may not fully capture, such as measurement noise, latent confounding, or domain-specific constraints. In our study, we focus on a fundamental property of distributions induced by causal graphs—specifically, the cancellation of paths phenomenon (see [StrongFaith]). This property has been previously analyzed in linear systems, and we extend this evaluation to nonlinear (NN-based) functions. While real-world datasets may differ from synthetic ones in various ways, this phenomenon is not tied to a specific functional class, noise type, or error but rather emerges from the structural properties of the graph itself. Therefore, we argue that it is likely to be relevant in many real-world settings.
> > >
> > > That said, we acknowledge that empirical validation on real-world datasets is essential for assessing the practical impact of these findings.
> > >
> > > **References**
> > >
> > > [DCDI] BROUILLARD, Philippe, et al. Differentiable causal discovery from interventional data. NeurIPS 2020.
> > >
> > > [BayesDag]ANNADANI, Yashas, et al. Bayesdag: Gradient-based posterior inference for causal discovery. NeurIPS 2023.
> > >
> > > [GraN-Dag] LACHAPELLE, Sébastien, et al. Gradient-based neural dag learning. arXiv preprint, 2019.
> > >
> > > [CAM] BÜHLMANN, Peter; PETERS, Jonas; ERNEST, Jan. CAM: Causal additive models, high-dimensional order search and penalized regression. 2014.
> > >
> > > [SDCD] NAZARET, Achille, et al. Stable differentiable causal discovery. ICML 2024.
> > >
> > > [DiscoGEN] KE, N. R., et al. DiscoGen: Learning to Discover Gene Regulatory Networks. 2023.
> > >
> > > [AVICI] LORCH, Lars, et al. Amortized inference for causal structure learning. NeurIPS 2022.
> > >
> > > [FIP] SCETBON, Meyer, et al. A Fixed-Point Approach for Causal Generative Modeling. ICML 2024.
> > >
> > > [Grounding] BROUILLARD, Philippe, et al. The Landscape of Causal Discovery Data: Grounding Causal Discovery in Real-World Applications. arXiv preprint, 2024.
> > >
> > > [StrongFaith] ZHANG, Jiji; SPIRTES, Peter. Strong faithfulness and uniform consistency in causal inference. UAI 2002.

---

### Official Review · Reviewer_6UaB · 2025-03-14

**Overall Recommendation:** 4

**Summary:**

This work critically examines the limitations of neural causal discovery methods, revealing their fundamental inability to distinguish causal relationships in finite-sample regimes reliably. Through a systematic benchmarking protocol, the authors demonstrate that even state-of-the-art neural approaches struggle to recover ground-truth causal structures, attributing this failure to estimation errors and the violation of the faithfulness assumption. The study quantifies the difficulty of causal discovery using the λ-strong faithfulness property and shows that as graph size increases, the proportion of faithful distributions decreases exponentially, fundamentally constraining current methods.

**Claims And Evidence:**

The paper provides strong empirical evidence to support its claims, using a rigorous controlled experiments. The claim that neural causal discovery methods struggle to recover ground-truth structures is well-supported by systematic evaluations across multiple datasets and methods. Additionally, the argument that the faithfulness assumption is a key bottleneck is backed by quantitative analysis of λ-strong faithfulness.

**Essential References Not Discussed:**

None

**Experimental Designs Or Analyses:**

- As mentioned previously, this work focuses only on synthetic data, while recommending others, in the discussion section, to use real-world datasets.
- The graphs analyzed in this work were generated only using the Erdos-Renyi model. It would be insightful to know whether the results generalizes to other graph classes such as scale-free, small-world, etc.
- Additionally, authors can perform ablation studies to better understand how different neural network architectures (depth, width, activation functions) affect causal graph recovery.

**Methods And Evaluation Criteria:**

The paper introduces a systematic benchmarking framework that standardizes datasets, hyperparameter tuning, and functional approximations, ensuring robust and fair comparisons across methods. The use of synthetic datasets with known ground-truth causal structures is an appropriate choice for evaluating structural recovery accuracy, while the incorporation of the λ-strong faithfulness property provides a theoretically grounded measure of dataset difficulty. This approach effectively demonstrates the limitations of existing neural methods. However, the inclusion of real-world datasets would further strengthen the study by assessing whether these challenges persist in practical applications.

**Other Comments Or Suggestions:**

Please correct the following typos:
- Section 1 (line 48): unfaithfull -> unfaithful
- Section 2, the line “An SCM defines a joint distibution P over the set of random vairables {Xi}” is repeated twice.
- Multiple occurrences of “distibution” instead of “distribution.”
- Figure 1, last sentence has incorrect section reference.
- Title of section 3 and 3.1 are same.
- Section 3.2 (line 245): hat -> that
- Section 4 (line 258) -> ot -> to

**Other Strengths And Weaknesses:**

The paper is generally well-written, aside from the following typos.

**Questions For Authors:**

None

**Relation To Broader Scientific Literature:**

While neural methods aim to improve scalability, the study empirically shows they struggle under finite samples due to faithfulness violations. By quantifying the impact of λ-strong faithfulness, the paper highlights fundamental constraints and the need for alternative causal discovery paradigms.

**Theoretical Claims:**

This work primarily focuses on empirical contributions rather than formal theoretical proofs.

---

> ### Author Rebuttal · Authors · 2025-04-01
>
> We sincerely appreciate the reviewer’s thoughtful feedback and their positive evaluation of our work. We are especially grateful for the recognition of the strong empirical evidence supporting our claims and rigorous controlled experiments that validate our approach. It is encouraging to hear that our systematic evaluations effectively support our findings and that our methodology successfully highlights the limitations of existing neural methods.
>
>
> Additionally, we appreciate the acknowledgment that the use of synthetic datasets with known ground-truth causal structures is an appropriate choice. Our goal was to ensure a clear and interpretable evaluation, and we are pleased that the reviewer found this aspect of our work well justified. Finally, we are glad that the reviewer found the paper to be generally well-written, and we thank them for their time and effort in assessing our work.
>
>
> Regarding suggestions from *Experimental Designs Or Analyses*:
>
> 1. We agree that evaluations on real-world graphs are vital for monitoring progress in causal discovery. However, there is a very limited amount of real-world data annotated with ground-truth graphs. Thus, the only viable way to conduct a comprehensive, large-scale analysis is to resort to synthetic data. Moreover,  we follow the standard practice in causal discovery research by using widely accepted synthetic benchmarks. These datasets are not designed adversarially but are generated randomly. Thus, we believe the results are generalizable to real-world datasets.
>
> 2. Further, during the rebuttal phase, we conducted additional analyses of the lambda-faithfulness property for other types of graphs (scale-free, small-world, and bipartite). The results align with the observations on ER graphs, see Figure R.1 in the [rebuttal material](https://drive.google.com/file/d/1m7rFfvf2_xoQorCprk6RE1MiITmYivhy/view?usp=sharing). These results will be added to the final version of the paper. We hope the reviewer finds them interesting and reassuring.
>
> 3. During the project, we conducted multiple ablations regarding the neural network architectures. Some of them are described in the appendix. Table 9 in Appendix A analyzes the impact of the network size on the performance of our optimized algorithm introduced in Section 3. In Appendix B, we provide a detailed study of the influence of network architecture on the performance of selected neural causal discovery methods. Notably, we compared architectures with residual connections and with layer norm; see Figure 9 in Appendix B. In all the cases, we found the behavior very similar to the one presented in the main body of the paper. We add a remark stating this.
>
> We hope that the above answers resolve the reviewer’s concern. However, we’d be happy to perform additional analysis and add clarifications should the reviewer find that something is still missing.

---

### Decision · Program_Chairs · 2025-05-01

**Decision:**

Accept (poster)

**Comment:**

The idea of this work is simple and is based on the premise that the strong assumption of faithfulness is responsible for the non-improvements in the neural causal discovery. The authors demonstrate that even state-of-the-art neural approaches struggle to recover ground-truth causal structures. To this end, a benchmark is proposed to evaluate several discovery algorithms. Additionally, a metric for measuring the degree of faithfulness is introduced to assess how faithfulness impacts the performance of these algorithms.

Overall, the paper received 3 reviews with the majority of the reviewers assessing the work positively. The critical reviewer had questions regarding the generalizability of the faithfulness assumption bottleneck along with how well the bottleneck holds when the data is scaled. The AC read the paper, and have certain questions that need to be clarified in the paper:

1. NN-Opt is a brute force method, so I have serious concerns about the efficiency of the method.

2. As reviewer 6UaB also points out, the experiments are only on synthetic data sets, but the whole motivation of the work is based on how this assumption is a problem in real-world scenarios.

3. I also agree with Reviewer JYG6, that 80k examples are not many. If the authors are focusing on synthetic examples, nothing prevents them from increasing the size of the data set. There are works such as Ribeiro-Dantas et al., Learning interpretable causal networks from very large datasets: application to 400,000 medical records of breast cancer patients, iScience 2024 and Lopez et al., Large-Scale Differentiable Causal Discovery of Factor Graphs, NeurIPS 2024

4. I also have a question regarding Fig 3 b) SDCD and Bayes DAF show a downward trend, so I think the claim that there is no improvement in the ESHD metric is a strong one. I believe the authors should conduct more evaluations on this.

Overall, I think this is a good paper, but there are several questions that are left unanswered. I recommend a weak acceptance.